# Tissue-specific *O*-GlcNAcylation profiling identifies substrates in translational machinery in *Drosophila* mushroom body contributing to olfactory learning

Haibin Yu[1], Dandan Liu[2], Yaowen Zhang[1], Ruijun Tang[1], Xunan Fan[1], Song Mao[1], Lu Lv[1], Fang Chen[1], Hongtao Qin[3], Zhuohua Zhang[1,4], Daan MF van Aalten[5], Bing Yang[2], Kai Yuan[1,4,6]*

[1]Hunan Key Laboratory of Molecular Precision Medicine, Department of Oncology, Xiangya Hospital & Center for Medical Genetics, School of Life Sciences, Central South University, Changsha, China; [2]Life Sciences Institute, Zhejiang University, Hangzhou, Zhejiang, China; [3]State Key Laboratory of Chemo/Biosensing and Chemometrics, College of Biology, Hunan University, Changsha, China; [4]National Clinical Research Center for Geriatric Disorders, Xiangya Hospital, Central South University, Changsha, China; [5]Department of Molecular Biology and Genetics, University of Aarhus, Aarhus, Denmark; [6]The Biobank of Xiangya Hospital, Central South University, Changsha, China

*For correspondence:
yuankai@csu.edu.cn

Competing interest: The authors declare that no competing interests exist.

**Abstract** *O*-GlcNAcylation is a dynamic post-translational modification that diversifies the proteome. Its dysregulation is associated with neurological disorders that impair cognitive function, and yet identification of phenotype-relevant candidate substrates in a brain-region specific manner remains unfeasible. By combining an *O*-GlcNAc binding activity derived from *Clostridium perfringens* OGA (*Cp*OGA) with TurboID proximity labeling in *Drosophila*, we developed an *O*-GlcNAcylation profiling tool that translates *O*-GlcNAc modification into biotin conjugation for tissue-specific candidate substrates enrichment. We mapped the *O*-GlcNAc interactome in major brain regions of *Drosophila* and found that components of the translational machinery, particularly ribosomal subunits, were abundantly *O*-GlcNAcylated in the mushroom body of *Drosophila* brain. Hypo-*O*-GlcNAcylation induced by ectopic expression of active *Cp*OGA in the mushroom body decreased local translational activity, leading to olfactory learning deficits that could be rescued by dMyc overexpression-induced increase of protein synthesis. Our study provides a useful tool for future dissection of tissue-specific functions of *O*-GlcNAcylation in *Drosophila*, and suggests a possibility that *O*-GlcNAcylation impacts cognitive function via regulating regional translational activity in the brain.

## Editor's evaluation

This valuable study provides solid evidence that within the *Drosophila* brain there are regionally regulated patterns of O-linked modification of proteins with the monosaccharide N-Acetyl glucosamine. Using a novel and powerful method of identifying proteins bearing this modification, the authors provide evidence that in a region of the *Drosophila* brain critical for associative learning, the mushroom body, there is a high representation of modified proteins affecting protein translation. Reductions in GlcNAc modification affects both an associative learning task and new protein synthesis, suggesting a critical function of these monosaccharide modifications in the regulation of

protein synthesis required for memory formation. These findings provide a putative mechanism for human neurological deficits that accompany reductions in this ubiquitous carbohydrate modification.

## Introduction

Protein O-GlcNAcylation is a ubiquitous post-translational modification that occurs on thousands of nuclear and cytoplasmic proteins, conveying various stimuli or stressors such as fluctuating nutrient levels to distinct cellular processes (*Yang and Qian, 2017*; *Olivier-Van Stichelen and Hanover, 2015*; *Davie et al., 2018*). It involves reversible attachment of β-*N*-acetylglucosamine (GlcNAc) to the hydroxyl group of serine and threonine residues of protein substrates, catalyzed by a pair of evolutionarily conserved enzymes, O-GlcNAc transferase (OGT) and O-GlcNAcase (OGA) (*Vocadlo, 2012*). As a monosaccharide modification, the addition and removal of O-GlcNAc moiety are dynamic, with cycling rates ranging from several minutes to the lifetime of a protein (*Miller et al., 1999*; *Roquemore et al., 1996*). By modifying different protein substrates, O-GlcNAcylation exerts critical regulatory functions in a wide range of basic cellular processes, including transcription, translation, and protein homeostasis (*Yang and Qian, 2017*; *Uygar and Lagerlöf, 2023*; *Wang et al., 2023*). O-GlcNAcylation is ubiquitously distributed but more abundant in some tissues, such as the brain (*Fehl and Hanover, 2022*; *Wulff-Fuentes et al., 2021*). Given its enrichment in brain tissues and essential biological functions, it is not surprising that O-GlcNAc cycling is required for the development and functions of central nervous system (*Olivier-Van Stichelen and Hanover, 2015*; *Lagerlöf, 2018*; *Akan et al., 2018*), and its dysregulation is linked to numerous neurological disorders (*Uygar and Lagerlöf, 2023*; *Lagerlöf, 2018*; *Lee et al., 2021*; *Banerjee et al., 2016*).

O-GlcNAc homeostasis appears to be required for proper cognitive function, although the molecular connections between the dysregulated O-GlcNAcome and cognitive impairment are not fully understood. Hypomorphic mutations of *OGT* are implicated in an X-linked intellectual disability syndrome (*Pravata et al., 2020a*; *Pravata et al., 2019*; *Selvan et al., 2018*; *Willems et al., 2017*; *Vaidyanathan et al., 2017*), a severe neurodevelopmental disorder now termed *OGT*-associated Congenital Disorder of Glycosylation (OGT-CDG) (*Pravata et al., 2020b*). *Drosophila* models of OGT-CDG that carry the equivalent human disease-related *OGT* missense mutations manifest deficits in sleep and habituation, an evolutionarily conserved form of non-associative learning (*Fenckova et al., 2022*). Our recent work has shown that decreased O-GlcNAcylation level in *Drosophila*, induced through overexpression of a bacterial OGA from *Clostridium perfringens* (*Cp*OGA), leads to a deficit of associative olfactory learning. More interestingly, ectopic expression of *Cp*OGA during early embryogenesis results in reduced brain size and learning defects in adult flies, likely due to interference of the sog-Dpp signaling required for neuroectoderm specification (*Zhang et al., 2023*). These studies reveal that disturbed O-GlcNAc homeostasis can impact cognitive function by compromising neuronal development. On the other hand, a number of studies have revealed that impaired O-GlcNAcylation is implicated in aging-related neurodegenerative diseases such as Alzheimer's disease (AD) (*Uygar and Lagerlöf, 2023*; *Lagerlöf, 2018*; *Lee et al., 2021*; *Banerjee et al., 2016*; *Balana and Pratt, 2021*; *Quan et al., 2023*). In the cerebrum of AD patients, O-GlcNAcylation levels are significantly lower than that of healthy controls (*Liu et al., 2009*). Upregulation of O-GlcNAcylation levels by limiting OGA activity recovers the impaired cognitive function in AD mice models (*Park et al., 2021*; *Kim et al., 2013*). Interestingly, during normal aging in mice, reduction of O-GlcNAcylation levels also occurs in the hippocampus, and elevation of neuronal O-GlcNAc modification ameliorates associative learning and memory (*Wheatley et al., 2019*). These results indicate that, in addition to its involvement in neurodevelopment, O-GlcNAc homeostasis is also required for normal neuronal activity and cognitive function. However, the identity of key O-GlcNAc protein substrates supporting the cognitive abilities in adult brain and their spatial distribution remain largely unknown.

An obstacle to comprehensively identifying the O-GlcNAc conveyors of various cognitive functions is the lack of an effective tissue-specific O-GlcNAc profiling method. Given the structural diversity and relatively low abundance, enrichment of O-GlcNAc-modified proteins is required for mass spectrometry (MS)-based profiling of O-GlcNAcylation (*Yin et al., 2021*). The enrichment strategies roughly fall into two categories. One category involves direct capture of O-GlcNAcylated proteins by antibodies or lectins that recognize the GlcNAc moiety (*Yin et al., 2021*; *Hu et al., 2022*; *Dupas et al., 2022*; *Saha et al., 2021*; *Maynard and Chalkley, 2021*; *Ma et al., 2021a*; *Ma et al., 2021b*).

**eLife digest** Newly synthesized proteins often receive further chemical modifications that change their structure and role in the cell. *O*-GlcNAcylation, for instance, consists in a certain type of sugar molecule being added onto dedicated protein segments. It is required for the central nervous system to develop and work properly; in fact, several neurological disorders such as Alzheimer's, Parkinson's or Huntington's disease are linked to disruptions in *O*-GlcNAcylation. However, scientists are currently lacking approaches that would allow them to reliably identify which proteins require *O*-GlcNAcylation in specific regions of the brain to ensure proper cognitive health.

To address this gap, Yu et al. developed a profiling tool that allowed them to probe *O*-GlcNAcylation protein targets in different tissues of fruit flies. Their approach relies on genetically manipulating the animals so that a certain brain area overproduces two enzymes that work in tandem; the first binds specifically to *O*-GlcNAcylated proteins, which allows the second to add a small 'biotin' tag to them. Tagged proteins can then be captured and identified.

Using this tool helped Yu et al. map out which proteins go through *O*-GlcNAcylation in various brain regions. This revealed, for example, that in the mushroom body – the 'learning center' of the fly brain – *O*-GlcNAcylation occurred predominantly in the protein-building machinery.

To investigate the role of *O*-GlcNAcylation in protein synthesis and learning, Yu et al. used an approach that allowed them to decrease the levels of *O*-GlcNAcylation in the mushroom body. This resulted in reduced local protein production and the flies performing poorly in olfactory learning tasks. However, artificially increasing protein synthesis reversed these deficits.

Overall, the work by Yu et al. provides a useful tool for studying the tissue-specific effects of *O*-GlcNAcylation in fruit flies, and its role in learning. Further studies should explore how this process may be linked to cognitive function by altering protein synthesis in the brain.

*O*-GlcNAc antibodies including RL2 and CTD110.6, as well as *O*-GlcNAc-binding lectins such as wheat germ agglutinin (WGA), are commonly used for enrichment. In addition, the catalytic-dead mutant of *Cp*OGA that retains the ability to recognize *O*-GlcNAcylated substrates was successfully repurposed to concentrate many developmental regulators from *Drosophila* embryo lysates (*Selvan et al., 2017*). Another category of enrichment strategies relies on chemoenzymatic or metabolic labeling (*Yin et al., 2021*; *Hu et al., 2022*; *Dupas et al., 2022*; *Saha et al., 2021*; *Maynard and Chalkley, 2021*; *Ma et al., 2021a*; *Ma et al., 2021b*). Azido-modified intermediates, such as *N*-azidoacetylglucosamine (GlcNAz) and *N*-azidoacetylgalactosamine (GalNAz), are used to introduce specific tags (e.g. biotin) to protein substrates via Staudinger ligation or click chemistry, allowing for capture and enrichment of *O*-GlcNAcylated proteins. A recent study coupled the *O*-GlcNAc-binding lectin GafD to the proximity labeling TurboID yielding the GlycoID tool (*Liu et al., 2022*), in which GafD domain recognizes *O*-GlcNAcylated substrates and the TurboID enzyme attaches nonhydrolyzable biotin tags to proximal proteins within approximately 10 nm radius (*Branon et al., 2018*). The GlycoID tool was used to profile *O*-GlcNAcylation in different subcellular spaces including the nucleus and cytosol (*Liu et al., 2022*). It is noteworthy that the *O*-GlcNAcylated proteins identified by different *O*-GlcNAcylation profiling strategies are quite diverse, probably due to the dynamic nature of *O*-GlcNAc cycling as well as the potential bias in substrates preference intrinsic to the methods (*Zachara et al., 2004*; *Ma and Hart, 2014*). Nonetheless, these advancements have greatly expanded the pan-*O*-GlcNAcome over the past 30 years (*Wulff-Fuentes et al., 2021Ma et al., 2021a*). However, none of them has been adopted for tissue-specific identification of *O*-GlcNAcylated proteins.

Here, we generated transgenic *Drosophila* lines that allow specific expression of *Cp*OGA in different brain regions. Ectopic expression of *Cp*OGA in the major learning center of *Drosophila* brain, the mushroom body, reduced local *O*-GlcNAcylation levels and impaired olfactory learning. We further combined a catalytically incompetent *Cp*OGA mutant (*Cp*OGA[CD]) with the proximity labeling enzyme TurboID to develop an *O*-GlcNAcylation profiling tool. By conditional expression of this tool to translate *O*-GlcNAc modification into biotin conjugation in specific brain structures, we mapped the *O*-GlcNAc interactome and generated an *O*-GlcNAc atlas for different brain regions of *Drosophila* (tsOGA, https://www.kyuanlab.com/tsOGA). Particularly, we detected abundant *O*-GlcNAc modifications associated with protein components of the translational machinery in the mushroom body.

Lowering the mushroom body *O*-GlcNAcylation levels reduced the synthesis of new proteins, interfering with olfactory learning, which could be reversed by increasing ribosomal biogenesis via overexpression of dMyc. We propose that compromised translational activity in the brain learning center contributes to the cognitive deficits of *O*-GlcNAcylation insufficiency-associated neurological diseases.

## Results

### Perturbation of the mushroom body *O*-GlcNAcylation leads to olfactory learning deficits

We previously reported that ubiquitous expression of *Cp*OGA in *Drosophila* reduced global *O*-GlcNAcylation levels and resulted in impaired olfactory learning (*Zhang et al., 2023*). To determine which brain region was responsible for this hypo-*O*-GlcNAcylation induced learning defect, we conditionally expressed wild-type *Cp*OGA (*Cp*OGA^WT) in different brain structures of *Drosophila* (*Figure 1A*). *Cp*OGA^DM, which carries two point-mutations (D298N and D401A) that inactivate both the catalytic and binding activities toward *O*-GlcNAc modification, was used as a control. We dissected brains from the adult flies and validated tissue-specific expression patterns via immunostaining. As expected, Elav-Gal4 induced *Cp*OGA^WT expression in the whole brain (*Figure 1B*), leading to decreased *O*-GlcNAcylation levels compared to the *Cp*OGA^DM (*Figure 1C*). Similarly, other tissue-specific Gal4 drivers activated *Cp*OGA expression in different brain structures and perturbed local *O*-GlcNAc modifications. For instance, OK107-Gal4 drove *Cp*OGA^WT expression in the mushroom body and downregulated *O*-GlcNAcylation levels in the Kenyon cells (*Figure 1D and E*).

We then evaluated the cognitive ability of these flies using an olfactory learning assay as previously reported (*Jia et al., 2021*; *Mariano et al., 2020*; *Busto et al., 2010*). To rule out the possibility that overexpression of *Cp*OGA^WT or *Cp*OGA^DM differentially disrupted odor preference, we tested their olfactory acuity toward either 4-methylcyclohexanol (MCH) or octanol (OCT) using air as a control. Tissue-specific expression of *Cp*OGA^WT or *Cp*OGA^DM in the antennal and optic lobes caused differences in odor susceptibility toward MCH or OCT, and these flies were, therefore, not included in subsequent olfactory learning tests (*Figure 1—figure supplement 1A* and B). Flies expressing *Cp*OGA^WT or *Cp*OGA^DM in brain neurons, mushroom body, or ellipsoid body were trained to associate electric shock punishment with an air current containing MCH or OCT, and then tested for the ability to remember the electric shock-associated odor using a T-maze apparatus (*Figure 1—figure supplement 1C*). Compared to *Cp*OGA^DM, conditional expression of *Cp*OGA^WT in brain neurons or mushroom body compromised the ability to establish the association between odor and electric shock (*Figure 1F*), suggesting that decreased *O*-GlcNAcylation levels in these brain regions resulted in a deficit in olfactory learning. In contrast, flies expressing *Cp*OGA^WT or *Cp*OGA^DM in the ellipsoid body, as well as the control flies without a Gal4 driver, showed no statistical difference in the learning performance (*Figure 1F*). Ectopic expression of *Cp*OGA^WT in the mushroom body driven by OK107-Gal4 might impact neuronal development during the larval stages (*Zhang et al., 2023*). To directly investigate whether perturbation of *O*-GlcNAcylation compromised neuronal function in adult flies, we used the temperature-sensitive Gal80 (Gal80^ts) to restrict *Cp*OGA expression until adulthood (*Figure 1—figure supplement 1D*). This temporally controlled expression of *Cp*OGA^WT specifically in the adult mushroom body did not affect the odor acuity but significantly disrupted olfactory learning relative to *Cp*OGA^DM control (*Figure 1G*, *Figure 1—figure supplement 1A* and B). These results suggested that proper *O*-GlcNAcylation homeostasis is essential for the mushroom body function.

### *O*-GlcNAcylation profiling through *Cp*OGA proximity labeling

The mushroom body is known to be the associative learning center in *Drosophila* brain (*Heisenberg, 2003*; *McGuire et al., 2001*). Having discovered that *O*-GlcNAcylation homeostasis in the mushroom body was critical for olfactory learning, we developed an *O*-GlcNAc profiling method that allows the identification of candidate *O*-GlcNAcylated protein substrates in this brain region. Mutation of the catalytic residue Asp298 to Asn (D298N) of *Cp*OGA (*Cp*OGA^CD) inactivates the enzymatic activity but retains its ability to bind *O*-GlcNAcylated peptides. Taking advantage of this property, far western, gel electrophoresis, proximity ligation, and imaging methods have been developed (*Selvan et al., 2017*; *Zhang et al., 2022*; *Song et al., 2021*; *Mariappa et al., 2015*; *Estevez et al., 2020*), and immobilized *Cp*OGA^CD has been successfully used to enrich *O*-GlcNAcylated substrates in vitro (*Selvan*

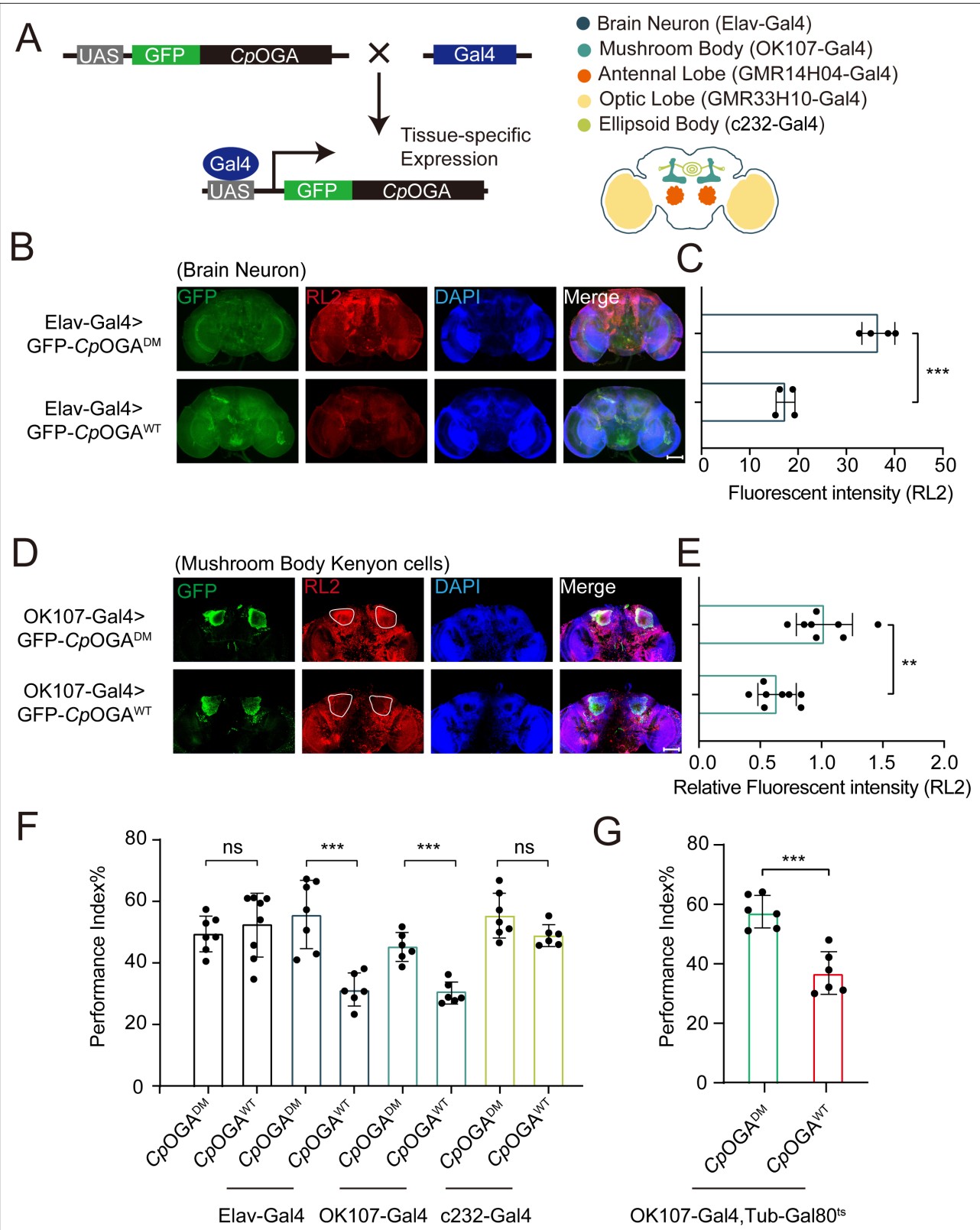

**Figure 1.** Downregulation of protein *O*-GlcNAcylation level in brain or mushroom body neurons affects olfactory learning of adult flies. (**A**) Scheme for expression of *Cp*OGA^WT or *Cp*OGA^DM in various *Drosophila* brain structures using different Gal4 drivers. (**B**) Immunostaining of adult *Drosophila* brains. Brains were stained with anti-*O*-GlcNAc antibody RL2 (red) to assess *O*-GlcNAcylation level, and anti-GFP (green) antibody to validate tissue-specific expression of *Cp*OGA. Nuclei were stained with DAPI (blue). Scale bar: 100 µm. (**C**) Quantification of fluorescent intensity of *O*-GlcNAc staining in

*Figure 1 continued on next page*

Figure 1 continued

*Cp*OGA$^{WT}$ or *Cp*OGA$^{DM}$ expressed brains. n = 4. (**D**) Immunostaining of adult *Drosophila* brains. Outlined areas indicate the cell bodies of Kenyon cells in mushroom body. Scale bar: 100 μm. (**E**) Quantification of relative fluorescent intensity of *O*-GlcNAc staining in *Cp*OGA$^{WT}$ or *Cp*OGA$^{DM}$ expressed brain structures. n = 8. (**F**) A compilation of performance index in learning test of the indicated flies expressing either *Cp*OGA$^{WT}$ or *Cp*OGA$^{DM}$. n = 6-8. (**G**) A compilation of learning performance index of flies expressing *Cp*OGA$^{WT}$ or *Cp*OGA$^{DM}$ only in the mushroom body at adult stage. n = 6. Each datapoint represents an independent experiment with approximately 200 flies. *p*-values were determined by unpaired *t*-test, and the stars indicate significant differences (\*\*\*p<0.001, \*\*p<0.01 and ns, not significant, p≥0.05). Error bars represent SD.

The online version of this article includes the following source data and figure supplement(s) for figure 1:

**Source data 1.** Excel spreadsheet containing source data used to generate *Figure 1C–G*.

**Figure supplement 1.** Impacts of reduction of *O*-GlcNAcylation in different brain structures on odor acuity towards 4-methylcyclohexanol (MCH) or octanol (OCT).

**Figure supplement 1—source data 1.** Excel spreadsheet containing source data used to generate *Figure 1—figure supplement 1A–B*.

*et al., 2017*). We linked this *O*-GlcNAc binding activity of *Cp*OGA$^{CD}$ with TurboID, a biotin ligase that catalyzes biotinylation of adjacent proteins (*Branon et al., 2018*), to tag the *O*-GlcNAcylated proteins with biotin for subsequent enrichment and Mass Spectrometry (MS) identification (*Figure 2A and B*). *Cp*OGA$^{DM}$ was adopted as a control to eliminate *O*-GlcNAc-independent protein-protein interactions (*Figure 2B*). Once induced by different tissue-specific drivers, this tool could tag and enrich *O*-GlcNAc substrates and their interactors in a tissue-specific manner, as endogenous protein biotinylation level is low in most organisms including *Drosophila*.

As proof of concept, we generated stable HEK293T cells expressing TurboID-*Cp*OGA$^{CD}$ or its reference construct TurboID-*Cp*OGA$^{DM}$. To characterize labeling activity, treatment with 10 mM or 100 mM biotin from an aqueous stock was first applied on these cells for 60 min, and the cell lysates were subject to western blot with streptavidin-HRP (*Figure 2—figure supplement 1A*). 10 mM biotin treatment yielded robust biotinylation of proteins, and this concentration was selected for subsequent experiments on cultured cells. To determine optimal incubation time, the cells were treated with 10 mM biotin from 15 to 180 min. Significant time-dependent labeling activity of proteins was observed, and 120 min was selected because it generated strong biotinylation in cells expressing *Cp*OGA$^{CD}$ compared to the *Cp*OGA$^{DM}$ control (*Figure 2—figure supplement 1B*). We validated whether a fluctuation in *O*-GlcNAcylation could be translated into biotinylation alterations. To this end, the cells were first treated with OGA inhibitor Thiamet-G or OGT inhibitor OSMI-1 for 6 hr followed by biotin incubation. Thiamet-G increased global *O*-GlcNAcylation levels, and the overall biotinylation was consistently upregulated. Conversely, OSMI-1 treatment decreased both *O*-GlcNAcylation and biotinylation in the cell lysates, suggesting that TurboID-*Cp*OGA$^{CD}$ effectively translates *O*-GlcNAc modification into biotin conjugation (*Figure 2—figure supplement 1C* and D).

To test whether TurboID-*Cp*OGA$^{CD}$ could be used to enrich and identify *O*-GlcNAcylated substrates, we performed immunoprecipitation with streptavidin magnetic beads from equal amount of cell lysates expressing either TurboID-*Cp*OGA$^{CD}$ or TurboID-*Cp*OGA$^{DM}$ after biotin incubation (*Figure 2C*). TurboID-*Cp*OGA$^{CD}$ labeled more proteins with biotin in the input compared to TurboID-*Cp*OGA$^{DM}$, and consistently, more biotinylated proteins were immunoprecipitated. Importantly, western blot with anti-*O*-GlcNAc antibody RL2 detected strong *O*-GlcNAcylation signals in immunoprecipitants from the cells expressing TurboID-*Cp*OGA$^{CD}$ but not TurboID-*Cp*OGA$^{DM}$, indicating successful enrichment of *O*-GlcNAc substrates using the biotin tags (*Figure 2C*). We scaled up the experiments and carried out MS analysis on the immunoprecipitants. Proteins that were selectively enriched in the TurboID-*Cp*OGA$^{CD}$ group relative to the TurboID-*Cp*OGA$^{DM}$ control ($\log_2$ FC >1) were regarded as *O*-GlcNAcylated substrates (*Figure 2B*). We, therefore, identified 336 *O*-GlcNAc candidate substrates from HEK293T cells (*Supplementary file 1*). To compare this result with known *O*-GlcNAc modifications, we compiled two lists of the previously identified *O*-GlcNAcylated proteins in HEK293T cells via either direct capture (*Zhao et al., 2011*; *Teo et al., 2010*) or chemoenzymatic labeling methods (*Wang et al., 2021*; *Zhu et al., 2020*; *Li et al., 2019*; *Li et al., 2016*; *Hahne et al., 2013*; *Supplementary file 2*). Gene ontology (GO) analysis on these three datasets showed that they were enriched in similar biological processes (*Figure 2—figure supplement 1E*). Overlap analysis revealed that 52% (178/336) of the *O*-GlcNAc candidate substrates identified in our study were also present in previous reports (*Figure 2D*). 48 proteins were shared among the three lists (*Supplementary file 3*), encompassing many well-known *O*-GlcNAcylated proteins such as OGT (*Griffin et al., 2016*), NUP153 (*Li et al.,*

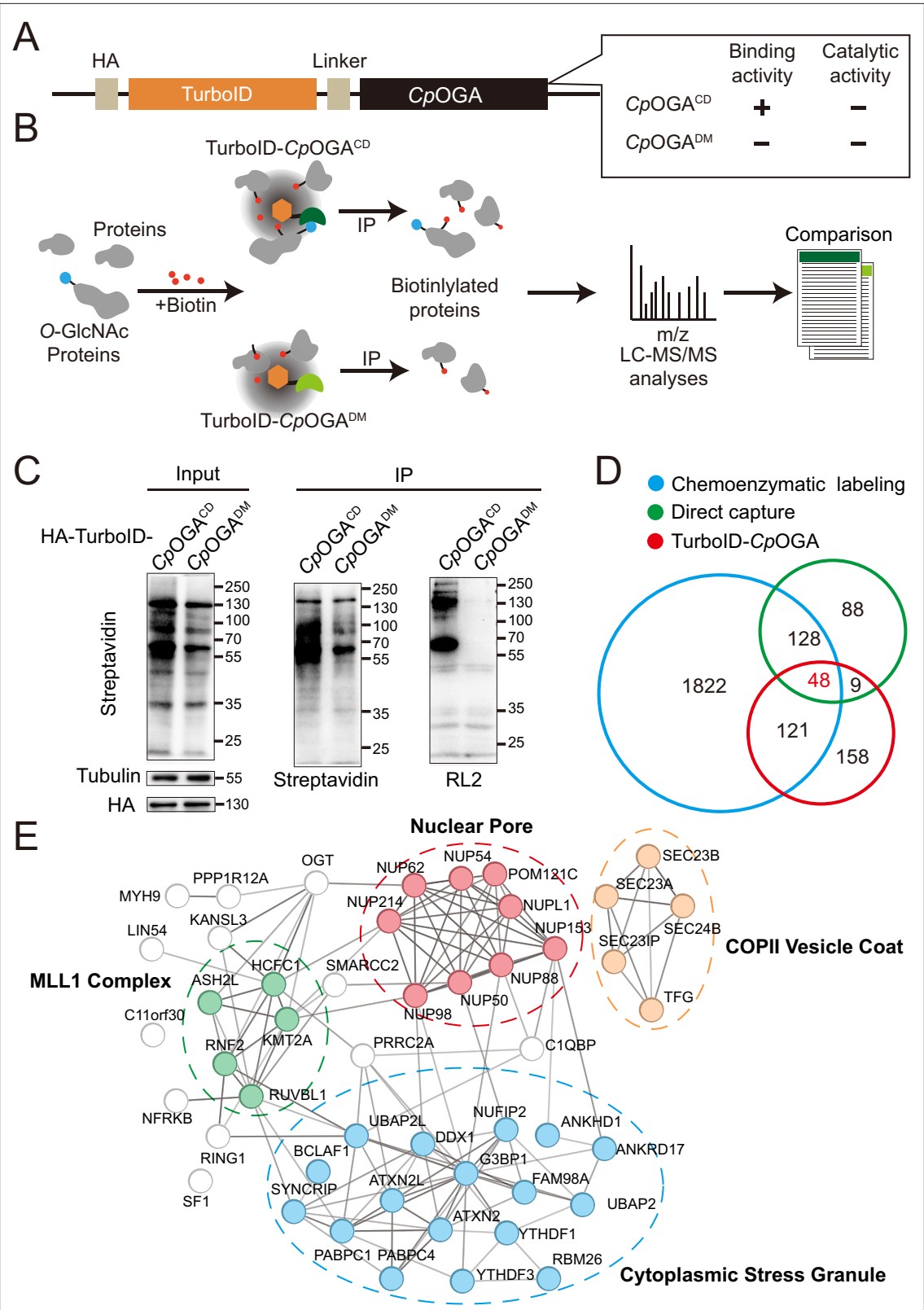

**Figure 2.** TurboID-*Cp*OGA^CD mediated proximity labeling of *O*-GlcNAc substrates in HEK293T cells. (**A**) Diagram of the constructs used for the expression of TurboID-*Cp*OGA^CD/DM. (**B**) Schematic representation of TurboID-*Cp*OGA^CD based profiling strategy. In the presence of biotin, TurboID biotinylates the *Cp*OGA^CD-bound *O*-GlcNAc proteins, which can be further purified by streptavidin pull-down for mass spectrometry (MS) identification. TurboID-*Cp*OGA^DM is used as a negative control for *O*-GlcNAc-independent protein-protein interactions. (**C**) Immunoprecipitation of biotinylated

*Figure 2 continued on next page*

*Figure 2 continued*

proteins from HEK293T cell lysates using streptavidin-magnetic beads. Biotinylation was detected by immunoblotting with streptavidin-HRP, and *O*-GlcNAcylation with anti-*O*-GlcNAc antibody (RL2). The expression of TurboID-*Cp*OGA[CD/DM] was verified by anti-HA immunoblotting. (**D**) Venn diagram showing the overlap of potentially *O*-GlcNAcylated proteins identified with TurboID-*Cp*OGA versus that with another two commonly used methods. (**E**) STRING visualization of protein-protein interaction network of the 48 highly-confident *O*-GlcNAc substrates in HEK293T cells.

The online version of this article includes the following source data and figure supplement(s) for figure 2:

**Source data 1.** Raw data of all western blots for *Figure 2*.

**Source data 2.** Complete and uncropped membranes of all western blots for *Figure 2*.

**Source data 3.** Excel spreadsheet containing source data used to generate *Figure 2C*.

**Figure supplement 1.** Validation and optimization of TurboID-*Cp*OGA[CD] mediated intracellular labeling.

**Figure supplement 1—source data 1.** Raw data of all western blots for *Figure 2—figure supplement 1*.

**Figure supplement 1—source data 2.** Complete and uncropped membranes of all western blots for *Figure 2—figure supplement 1*.

**Figure supplement 1—source data 3.** Excel spreadsheet containing source data used to generate *Figure 2—figure supplement 1A–E*.

---

*2022*), NUP62 (*Zhu et al., 2016*), and HCFC1 (*Capotosti et al., 2011*). Protein-protein interaction networks of these 48 proteins highlighted four cellular component clusters: the MLL1 complex, nuclear pores, COPII vesicle coats, and cytoplasmic stress granules (*Figure 2E*). Additionally, of the 158 candidate proteins that were unique in our result, 113 were annotated as *O*-GlcNAcylation substrates in the *O*-GlcNAc database (https://www.oglcnac.mcw.edu/). These results validated that TurboID-*Cp*OGA[CD] was able to selectively tag *O*-GlcNAcylated proteins with biotin for enrichment and identification.

## Region-specific *O*-GlcNAcylation profiling of *Drosophila* brain

We next generated transgenic flies harboring UAS-TurboID-*Cp*OGA[CD] or UAS-TurboID-*Cp*OGA[DM] via φC31 integrase-mediated site-specific recombination. To test biotinylation efficiency, we used Da-Gal4 to drive ubiquitous expression and raised the flies on biotin-containing food (100 mM) from early embryonic stage to adulthood according to previous reports (*Branon et al., 2018*; *Zhang et al., 2021*; *Figure 3A*). Flies were homogenized and equal amounts of lysate were used in immunoprecipitation experiments. Similar to the result with HEK293T cells, TurboID-*Cp*OGA[CD] catalyzed more biotinylation in the input relative to TurboID-*Cp*OGA[DM], and more biotinylated proteins were immunoprecipitated, in which strong *O*-GlcNAcylation signals were detected (*Figure 3B*). To validate whether TurboID-*Cp*OGA[CD] could achieve brain region-specific labeling of *O*-GlcNAcome with a biotin tag, we selected different Gal4 to drive TurboID-*Cp*OGA[CD] in distinct brain regions and fed the flies with biotin. Whole-mount staining of the brains showed that TurboID-*Cp*OGA[CD] displayed specific expression patterns as expected. More importantly, staining with streptavidin-Cy3 detected strong biotinylation in the brain regions expressing TurboID-*Cp*OGA[CD], whereas the rest of the brain showed negligible background signals (*Figure 3C*).

Subsequently, we immunoprecipitated biotinylated proteins from these fly brain lysates using streptavidin magnetic beads and performed MS analysis to identify putative *O*-GlcNAc substrates in different brain regions. Proteins with higher LFQ (label-free quantitation) intensity in the TurboID-*Cp*OGA[CD] group relative to the TurboID-*Cp*OGA[DM] control ($\log_2$ FC >1 or p<0.05) were considered as potentially *O*-GlcNAcylated substrates. We, therefore, identified 491 putative *O*-GlcNAcylated proteins in all neurons in the fly brain (Elav-Gal4), 455 in the mushroom body (OK107-Gal4), 377 in the antennal lobe (GMR14H04-Gal4), 234 in the optic lobe (GMR33H10-Gal4), and 289 in the ellipsoid body (c232-Gal4) (*Figure 3D*, *Supplementary files 4-8*). To obtain a functional overview of the *O*-GlcNAc interactome in different brain regions, GO analysis was performed to highlight the most enriched functional modules (*Figure 3E*, *Figure 3—figure supplement 1A–D*). The *O*-GlcNAc interactome in brain neurons was enriched in chemical synaptic transmission, neurotransmitter secretion, as well as chromatin remodeling, whereas putative *O*-GlcNAcylated substrates in specific brain regions were involved in rather diverse biological processes, ranging from mRNA splicing to chitin-base cuticle development. Of particular interest, putative *O*-GlcNAcylation modifications in the mushroom body were highly clustered in processes linked to translation, including cytoplasmic translation, translational initiation, ribosome assembly, and ribosome biogenesis. To eliminate possible interference caused by varying abundance of these candidate proteins in different brain regions, we normalized the calculated *O*-GlcNAc level ($\log_2$ FC) of each substrate using its corresponding brain region-specific normalizing

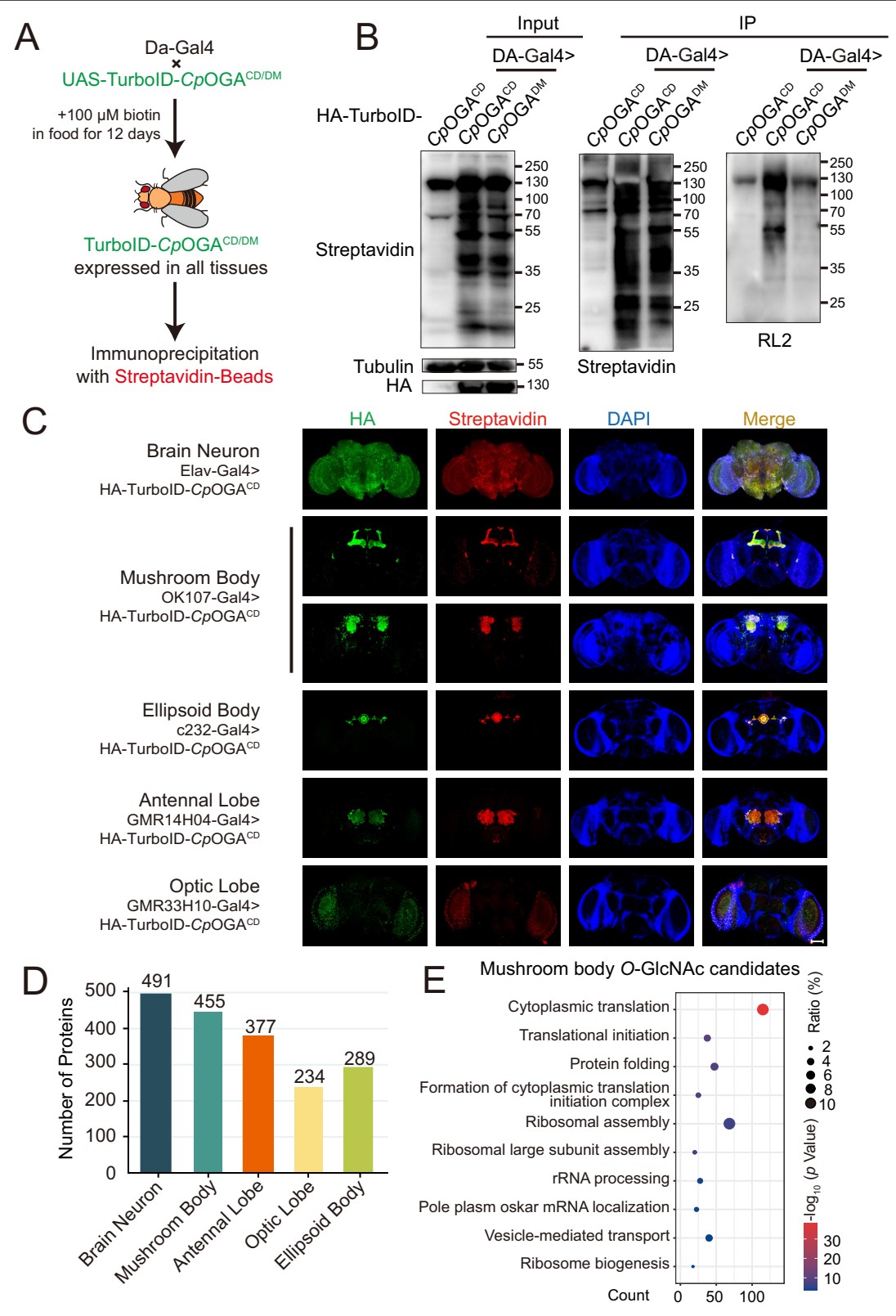

**Figure 3.** Identification of *O*-GlcNAc candidate substrates in different *Drosophila* brain structures using TurboID-Clostridium perfringens OGA (*Cp*OGA). (**A**) Scheme for validating TurboID-*Cp*OGA^CD/DM in flies. (**B**) Immunoprecipitation of biotinylated proteins from flies. Biotinylation was detected by immunoblotting with streptavidin-HRP, and *O*-GlcNAcylation with anti-*O*-GlcNAc antibody (RL2). The expression of TurboID-*Cp*OGA^CD/DM was validated by anti-HA immunoblotting. (**C**) Immunostaining of *Drosophila* brains expressing TurboID-*Cp*OGA^CD in different brain structures. Biotinylated

Figure 3 continued

proteins were stained with streptavidin-Cy3 (red), and TurboID-*Cp*OGA$^{CD}$ with anti-HA antibody. Nuclei were visualized by DAPI (blue). Scale bar: 100 μm. (D) Bar graph showing the number of *O*-GlcNAcylated protein candidates identified from different brain structures of *Drosophila*. (E) Gene Ontology (GO) enrichment analysis of *O*-GlcNAcylated protein candidates detected in the mushroom body. Bubble color indicates the -log$_{10}$ (*p*-value), and bubble size represents the ratio of genes in each category.

The online version of this article includes the following source data and figure supplement(s) for figure 3:

Source data 1. Raw data of all western blots for *Figure 3*.

Source data 2. Complete and uncropped membranes of all western blots for *Figure 3*.

Source data 3. Excel spreadsheet containing source data used to generate *Figure 3B–E*.

Figure supplement 1. GO analysis of candidate *O*-GlcNAc substrates from different brain regions of *Drosophila*.

Figure supplement 1—source data 1. Excel spreadsheet containing source data used to generate *Figure 3—figure supplement 1A–D*.

factor generated from the single-cell transcriptome atlas of the adult *Drosophila* brain (*Davie et al., 2018*; *Figure 3—figure supplement 1E*). For ease of search and use, we created an online database for tissue-specific *O*-GlcNAcylation Atlas of *Drosophila* Brain (tsOGA, http://kyuanlab.com/tsOGA/) to host these datasets (*Figure 3—figure supplement 1F*).

## *O*-GlcNAcylation affects cognitive function of *Drosophila* by regulating translational activity in the mushroom body

We calculated the percentage of ribosomal components in all the proteins identified from different brain regions, and found that nearly 10% of the putative *O*-GlcNAc substrates in the mushroom body were from ribosomes, much higher than that in other brain regions (*Figure 4—figure supplement 1A*). To validate that the observed enrichment was not due to higher expression levels of these ribosomal subunits in the mushroom body, we plotted the normalized *O*-GlcNAc levels of the putative ribosomal substrates alongside their mRNA abundances in different brain regions. While the *O*-GlcNAc levels were highest in the mushroom body, their mRNA abundances were not (*Figure 4A*). Moreover, in the mushroom body, the *O*-GlcNAc levels of these ribosomal proteins showed no correlation with their mRNA abundances (*Figure 4—figure supplement 1B*).

To directly verify whether mushroom body ribosomes were hyper-*O*-GlcNAcylated, Flag-tagged RPL13A, a core component of the large ribosomal subunit, was expressed in brain neurons or specifically in the mushroom body, driven by Elav-Gal4, or OK107-Gal4, respectively. Intact ribosomes were then isolated from these brain regions by anti-Flag immunoprecipitation (*Huang et al., 2019*; *Figure 4B*). Silver staining detected an array of specific bands on SDS-PAGE gel in the immunoprecipitants, indicating successful enrichment of ribosomal components. Western blot with anti-*O*-GlcNAc antibody RL2 showed that ribosomes purified from mushroom body contained more *O*-GlcNAc modifications than that from whole brain neurons. These results ascertained that ribosomal components were abundantly *O*-GlcNAc modified in the learning center of *Drosophila* brain.

To investigate whether high *O*-GlcNAcylation is required for translational activity in mushroom body, we dissected the brains of flies expressing *Cp*OGA$^{WT}$ driven by OK107-Gal4 and measured translation ex vivo using an O-propargyl-puromycin (OPP)-based protein synthesis assay (*Liu et al., 2012*; *Figure 4D*). Ectopic expression of *Cp*OGA$^{WT}$ but not the control *Cp*OGA$^{DM}$ in the mushroom body decreased local protein synthesis as visualized by the OPP fluorescent intensity (*Figure 4D and E*), suggesting that tuning down the *O*-GlcNAcylation compromised local translational activity. Hypo-*O*-GlcNAcylation in the mushroom body resulted in an olfactory learning defect (*Figure 1D and F*). We next investigated whether this cognitive phenotype was due to compromised translational activity. To this end, we selected a panel of representative ribosomal components that were significantly *O*-GlcNAcylated in the mushroom body, and performed RNA interference (RNAi)-mediated knockdown. The RNAi induced by Da-Gal4 reduced the expression of the targeted ribosomal genes to varying degrees (*Figure 4—figure supplement 1C*). We then crossed the RNAi lines to OK107-Gal4 to drive specific knockdowns in the mushroom body, and conducted an olfactory learning assay with these flies. Downregulation of RPL11 and RPL24 in the ribosomal large subunit, and RPS3 and RPS6 in the ribosomal small subunit did not alter olfactory acuity but led to compromised olfactory learning ability (*Figure 4—figure supplement 1D–F*), suggesting that reduction of translational activity was sufficient to cause learning impairment. We then reasoned that upregulation of translation might

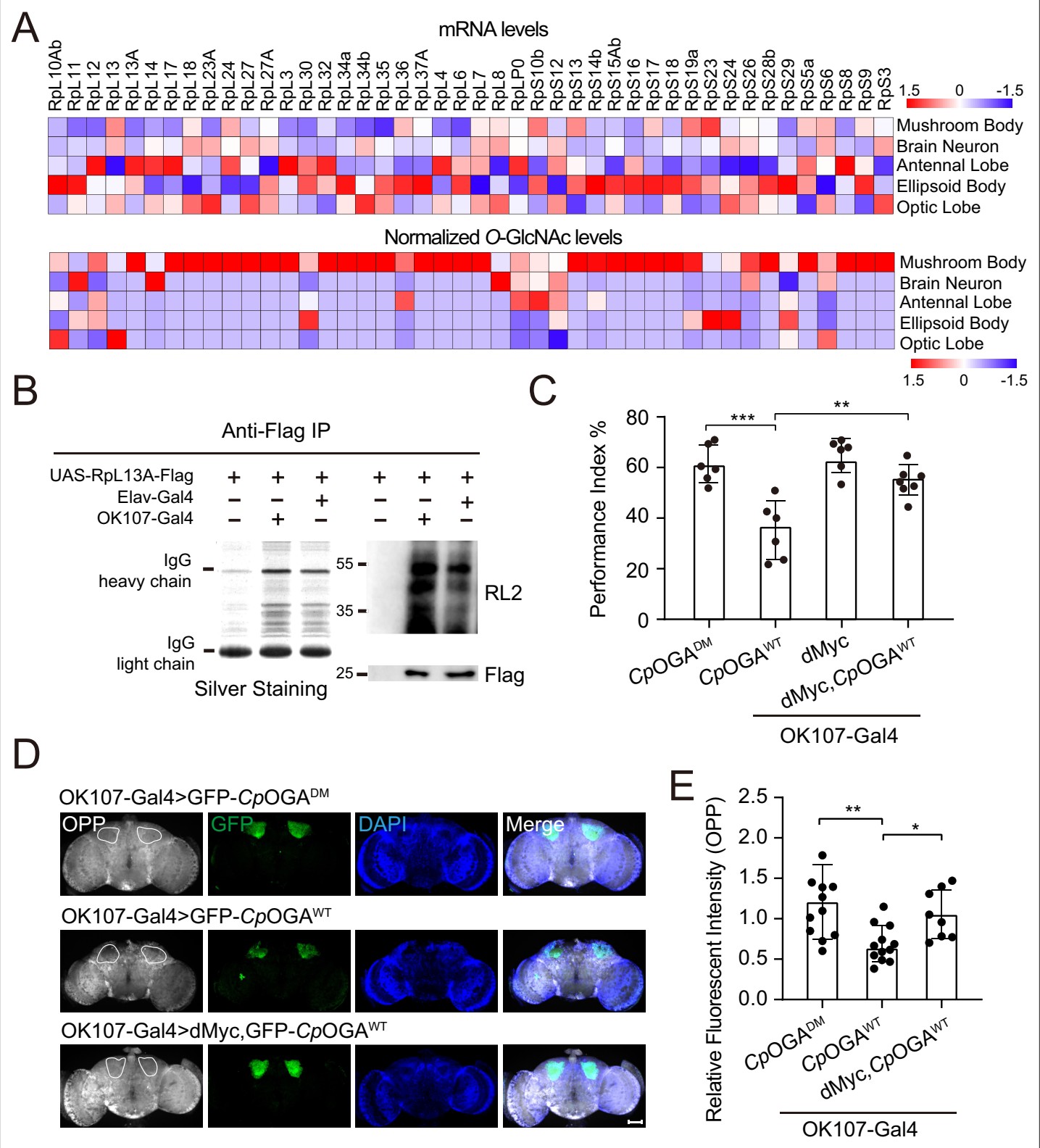

**Figure 4.** *O*-GlcNAcylation is required for proper protein synthesis activity and olfactory learning. (**A**) Heatmaps showing the mRNA levels (upper) and the normalized *O*-GlcNAc levels (lower) of the identified ribosomal candidates in different brain regions. (**B**) Immunoprecipitation of ribosomes using FLAG-tagged RpL13A. The expression of RpL13A-FLAG was validated by immunoblotting with anti-FLAG antibody. Ribosomal proteins were visualized using silver staining, and *O*-GlcNAcylation of ribosomes was analyzed by immunoblotting with anti-*O*-GlcNAc antibody RL2. (**C**) A compilation of the

*Figure 4 continued on next page*

*Figure 4 continued*

performance index of the indicated flies in the learning test. Learning defect of flies expressing *Cp*OGA[WT] was corrected by selective expression of dMyc in mushroom body. n = 6-7. Each datapoint represents an independent experiment with approximately 200 flies. (**D**) Ex vivo measurement of protein synthesis in mushroom body using the O-propargyl-puromycin (OPP) assay. Brains from the indicated flies were stained with anti-GFP (green) antibody to validate *Cp*OGA expression, and OPP (gray) to quantify protein synthesis. Nuclei were visualized with DAPI (blue). Outlined areas indicate the cell bodies of Kenyon cells of mushroom body. Scale bar: 100 μm. (**E**) Quantification of relative OPP fluorescent intensity in mushroom body regions. n = 8-12. *p*-values were determined by unpaired *t*-test, the stars indicate significant differences (***$p<0.001$, **$p<0.01$, *$p<0.05$). Error bars represent SD.

The online version of this article includes the following source data and figure supplement(s) for figure 4:

**Source data 1.** Raw data of all western blots for *Figure 4*.

**Source data 2.** Complete and uncropped membranes of all western blots for *Figure 4*.

**Source data 3.** Excel spreadsheet containing source data used to generate *Figure 4A–E*.

**Figure supplement 1.** Weakened ribosomal activity in mushroom body impacts olfactory learning.

**Figure supplement 1—source data 1.** Excel spreadsheet containing source data used to generate *Figure 4—figure supplement 1A–F*.

ameliorate the cognitive defect caused by *Cp*OGA[WT]-induced hypo-*O*-GlcNAcylation. Myc serves as a direct regulator of ribosome biogenesis, promoting protein synthesis through transcriptional control of RNA and protein components of ribosomes, as well as factors involved in the processing and nuclear export of these ribosomal subunits (*Gallant, 2013*; *Jiao et al., 2023*; *van Riggelen et al., 2010*). We overexpressed dMyc in the mushroom body to boost local translational activity. The results revealed that dMyc expression restored local protein synthesis, and more importantly, rescued the hypo-*O*-GlcNAcylation-induced olfactory learning defect (*Figure 4C–E*), indicating that *O*-GlcNAcylation insufficiency-induced cognitive impairment involves compromised translational activity in the brain learning center.

## Discussion

Protein *O*-GlcNAcylation is controlled by a very simple system consisting of only two enzymes, OGT and OGA. Yet it can dynamically modify more than 5000 protein substrates in different tissues to regulate their stability, protein-protein interactions, enzymatic activity, as well as subcellular localization upon changes in cellular metabolisms. Deciphering the spatial-temporal profiles of protein *O*-GlcNAcome and linking subsets of *O*-GlcNAc substrates to different physiological and pathological phenotypes are major obstacles in the field. In this study, we developed an *O*-GlcNAcylation profiling tool that allowed tissue-specific identification of *O*-GlcNAc candidate substrates. With this tool, we depicted the *O*-GlcNAc interactome in different brain regions of *Drosophila* and established an online database tsOGA (http://kyuanlab.com/tsOGA/) to facilitate future functional dissection of *O*-GlcNAcylation. Moreover, we consolidated a causal relationship between hypo-*O*-GlcNAcylation and cognitive impairment in *Drosophila*, and revealed that insufficient *O*-GlcNAcylation in the mushroom body of *Drosophila* brain reduced local translational activity that contributed to the observed olfactory learning deficits.

The *O*-GlcNAcome in different tissues and cell populations is heterogeneous and pleiotropic, and our understanding of the tissue-specific functions of *O*-GlcNAc modification remains quite limited, mainly relying on conditional knockout studies of *OGT* or *OGA* (*Issad et al., 2022*). Establishment of *O*-GlcNAcylation landscapes in different tissues under healthy and diseased conditions is needed to fully appreciate its multifaceted functions. The strategy reported here has achieved mapping the *O*-GlcNAcylated candidates with high spatial precision in *Drosophila* brain. With small modifications, this strategy can be readily applied to other tissues or even other model organisms in future studies. However, there are two potential caveats that need to be taken into consideration. First, the method relies on the ectopic expression of bacterial *Cp*OGA mutants fused with TurboID to label the *O*-GlcNAcome. The introduction of these foreign proteins could interfere with the normal functions of the targeted tissue. Although *Drosophila* seemed to tolerate this bacterial protein well when we assessed the functional consequences of expressing *Cp*OGA or its mutants in different tissues, their impacts on other model organisms remain unknown. Second, given that our method is based on the differential enrichment in the TurboID-*Cp*OGA[CD] experimental group relative to the TurboID-*Cp*OGA[DM] control group to identify putative *O*-GlcNAc substrates, the sensitivity is limited compared

to the chemoenzymatic labeling methods. Additionally, because the TurboID biotinylates all proximal proteins within ~10 nm radius, the identified proteins can be in complex with other *O*-GlcNAc substrates but itself is not directly *O*-GlcNAcylated. Further biochemical validations are needed to ascertain the *bona fide* substrates and their modification sites. Nonetheless, using the *O*-GlcNAc profiling data generated with this method, we established a framework of a tissue-specific *O*-GlcNAcylation database for *Drosophila*. As more tissue-specific *O*-GlcNAc profiling data are generated and deposited, it will undoubtedly be a useful resource for the community to facilitate future functional interrogations of different *O*-GlcNAcylation substrates at the organismal level.

The brain manifests high OGT expression and relies on protein *O*-GlcNAcylation to regulate many of its functions. Perturbed *O*-GlcNAcylation has been linked to neurodegenerative diseases and several key etiological factors are known *O*-GlcNAc substrates, such as tau (*Liu et al., 2009*; *Yuzwa et al., 2012*), β-amyloid (Aβ) (*Park et al., 2021*), neurofilaments (NFs) (*Lüdemann et al., 2005*), TDP-43 (*Zhao et al., 2021*), and α-synuclein (*Levine et al., 2019*; *Marotta et al., 2015*). Particularly, *O*-GlcNAcylation can antagonize hyperphosphorylation of tau and stabilize it from aggregation, preventing neuronal death and tauopathies (*Lee et al., 2021*). Hence, OGA inhibitors have been tested in several clinical trials to target tauopathy and early symptomatic AD, leading to a recent FDA approval of the OGA inhibitor MK-8719 as an orphan drug for tau-driven neurodegenerative disease (*Wang et al., 2020*). Our study strengthened a causal relationship between hypo-*O*-GlcNAcylation and cognitive impairment, and suggested that *O*-GlcNAcylation influences associative learning by regulating translational activity in the brain computational center. Consistent with previous reports (*Ohn et al., 2008*; *Shu et al., 2022*; *Zeidan et al., 2010*), we identified components in the translational machinery as putative *O*-GlcNAc substrates, including several translational initiation factors and particularly many ribosomal subunits. The potential regulation of ribosomal activity by *O*-GlcNAcylation warrants future structural and biochemical characterizations. Our *O*-GlcNAc profiling results also provide a rich resource for the discovery of other conveyors of *O*-GlcNAc-associated cognitive deficits. For instance, the brain *O*-GlcNAc substrates, scu and Upf3 possess human homologs, *HSD17B10*, and *UPF3B*, that are known X-linked intellectual disability risk genes (*Firth et al., 2009*; *Vissers et al., 2016*). In addition, recent studies have revealed that stress granules are tightly linked with autism spectrum disorders (*Jia et al., 2022*). The enrichment of stress granule components in the *O*-GlcNAc substrate list suggests that *O*-GlcNAcylation dysregulation might be involved in autism as well. We anticipate that this study will galvanize further studies into targeting *O*-GlcNAcylation insufficiency to ameliorate cognitive defects commonly seen in many neurological diseases.

## Materials and methods

### Cell cultures and generation of stable cell lines

HEK293T cells (Meisen CTCC) were cultured in a DMEM/high glucose medium (Biological Industries, 01-052-1A) with 10% FBS (VISTECH, SE100-B) at 37°C under 5% $CO_2$. The *Cp*OGA$^{CD}$ and *Cp*OGA$^{DM}$ sequences were codon optimized to *Homo sapiens* and *Drosophila* using *Jcat* (*Grote et al., 2005*). The fragments of *TurboID-CpOGA$^{CD}$* and *TurboID-CpOGA$^{DM}$* (*TurboID-CpOGA$^{CD/DM}$*) were PCR amplified and cloned into pCDH-CMV-HA vectors, respectively. For lentivirus preparation, HEK293T cells were transfected with *TurboID-CpOGA$^{CD/DM}$* plasmid with the packaging plasmids pPAX2 and pMD.2G using Polyethylenimine Linear (PEI, Polysciences, 24765). The PEI-containing medium was replaced with fresh serum-containing DMEM medium after 8 hr, and the viral supernatants were collected 48 hr and 72 hr post-transfection. The viral supernatants were centrifuged at 10,000 g for 1 hr at 4°C, and the pellet was dissolved in PBS (Biological Industries, 02-023-1A). HEK293T cells were infected in six-well plates and selected with 1 µg/mL Puromycin (Selleck, s7417) in the medium for at least 5 days. For biotin labeling, the TurboID-*Cp*OGA$^{CD}$ or TurboID-*Cp*OGA$^{DM}$ expressing HEK293T cells were labeled with 10–100 µM biotin (Merck, B4501) in the medium for 15 min to 3 hr. Labeling was stopped by placing cells on ice and washing cells three times with PBS (Biological Industries, 02-023-1A).

### *Drosophila* stocks and genetics

All flies were raised on standard fly food at 25 °C. Biotin food was prepared by adding 200 mM biotin (Merck, B4501) to hot (~60°C) standard fly food and dissolved to a final concentration of 100 µM (*Zhang et al., 2021*). The strains used in this study were as follows: *w1118,;sco/cyo;TM3/TM6B*,

*Da-Gal4* (Gift from Kun Xia's lab), *Elav-Gal4* (Gift from Zhuohua Zhang's lab), *OK107-Gal4, 201Y-Gal4* (Gift from Ranhui Duan's lab), *C232-Gal4* (BDSC, #30828), *GMR14H04-Gal4* (BDSC, #48655), *GMR33H10-Gal4* (BDSC, #49762), *Tub-Gal80^ts^, uas-RPL13A-FLAG, uas-dMyc* (Gift from Jun Ma's lab), *uas-shLuciferase* (Gift from Zhuohua Zhang's lab), *uas-shRPL5* (THU0670), *uas-shRPs26* (THU0747), *uas-shRPL24* (THU1411), *uas-shRPS6* (THU0864), *uas-shRPL11* (TH201500769.S), *uas-shRPS3* (THU1958), *uas-shRPL32* (TH201500773.S), *uas-shRPS28b* (THU1037). Our study established two transgenic fly lines (*UAS-HA-TurboID-CpOGA^CD^* and *UAS-HA-TurboID-CpOGA^DM^*). *TurboID-CpOGA^CD/DM^* fragments were cloned into pUASz-HS-HA vectors, respectively using Gibson assembly (NEB). Constructs with the attB sequence were injected into flies (*y1, w67c23; P(CaryP) attP2*) to initiate the φC31 integrase-mediated site-specific integration (UniHuaii). The resulted adult flies (G0) were crossed to double balancer to get the F1 generations.

## Olfactory learning and memory

Behavioral experiments were carried out in an environmental chamber at 25 °C and 70% humidity as previously described (*Jia et al., 2021*). We tested the acuity of flies against two aversive odors, 4-methylcyclohexanol (MCH, Sigma, 104191) and 3-octanol (OCT, Sigma, 218405). Approximately 100 flies were placed in the center compartment of the T-maze, where the collection tubes were snapped into place at the choice point and the air and aversive odor tubes were connected with the distal ends of the collection tubes. Flies were allowed to choose between air versus aversive odor for 2 min. After the choice period, the sliding center compartment was pulled up quickly, trapping the flies in the collection tubes they had chosen. Flies in each collection tube were anesthetized and counted. Performance index (PI^odor^) was determined as the number of flies on the air side (n(Air)) minus the number on the aversive odor side (n(odor)) divided by the total number of flies (n(Air)+n(odor)) and multiplied by 100%.

PI^odor^=[n(Air)-n(odor)]/[n(Air)+n(odor)]×100%.

If the experimental group flies have similar odor avoidance to that of control, they will be used for subsequent olfactory learning tests.

After confirming that the flies to be tested have avoidance behavior in response to electric shock, flies were trained to associate an aversive odor (MCH or OCT) used as a conditioned stimulus (CS) with electric shock. The experiment comprised two phases: the flies were trained in the first phase, and the trained flies were tested in the second phase. During training, approximately 100 flies were simultaneously exposed to odor 1 (CS^+^) and electric shock (60 V) in a training tube for 1 min. Then, they were exposed to the blank odor (air) for 1 min before receiving odor 2 (CS^-^) without electric shock for 1 min, followed by the blank odor (air) for 1 min. Immediately after training, flies were transferred to the central chamber of the T-maze and retained there for 1 min. To measure learning, The center chamber was slid smoothly into the register with the choice point of the T-maze and the MCH and OCT odor tubes were supplied from the two distal ends of the collection tube to let the flies choose between the two odors for 2 min. The central chamber then was pulled up quickly, trapping the flies in the collection tube they had chosen. Flies in each collection tube were anesthetized and counted. We calculated the Performance Index (PI) for each condition as the number of flies avoiding the shock-paired odor (CS^-^) minus the number of flies choosing the shock-paired odor (CS^+^) divided by the total number of flies (CS^-^ + CS^+^) and multiplied by 100%.

PI = [n(CS^-^)-n(CS^+^)]/[n(CS^+^)+n(CS^-^)]×100%.

In each experiment, we calculated the mean PI from two trials: one in which MCH was the shock-paired odor, and the other in which OCT was the shock-paired odor. This method removed any potential bias caused by the flies having a stronger preference for one odor over the other. Therefore, each point in the bar graph consisted of approximately 200 flies (male: female = 1:1), with half of the flies trained to one odor, and the other half trained to the other odor.

For the temporally controlled *Cp*OGA expression in the adult mushroom body, the flies were initially maintained at 19°C until adulthood. Then, the flies were transferred to 29°C for 3–5 days to inactivate Gal80^ts^ and hence allow the expression of *Cp*OGA. The behavioral experiments were carried out subsequently.

## Western blot assay

The HEK293T cells and flies were lysed in lysis buffer (2% SDS, 10% glycerol, and 62.5 mM Tris-HCl, pH 6.8) supplemented with protease inhibitor cocktail (1:100, Sigma, P8340), and PMSF (1:100, Sigma,

P7626) and 50 µM Thiamet-G (Selleck, s7213). Lysates were clarified by centrifugation at 13,000 rpm for 30 min at 4°C, and the protein concentration was determined using BCA assay (Beyotime, p0009). Proteins were mixed with an equal volume of SDS sample buffer (2% β-Mercaptoethanol) and boiled for 10 min at 95°C. Proteins were separated by 10% SDS-PAGE (90 V, 30 min; 120 V, 1 hr) and transferred to a Polyvinylidene Fluoride (PVDF, Millipore, IPVH00010) membrane (290 mA, 90 min). The PVDF membrane was blocked with 5% non-fat milk for 1 hr, then incubated with primary antibodies overnight at 4°C, and then incubated with secondary antibodies (1:5000, Thermo Fisher Scientific) for 1 hr at room temperature. The signal was detected using ECL substrates (Millipore). Primary antibodies were dissolved in 5% BSA (Biofroxx, 4240GR005) and the dilutions were: Streptavidin-HRP (1:2000, GenScript, M00091), RL2 (1:1000, Abcam, ab2739), HA (1:3000, Cell Signaling Technology, 3724), Tubulin (1:3000, Cell Signaling Technology, 12351 S), FLAG (1:3000, Cell Signaling Technology, 14793). For the Western blot experiment in *Figure 2—figure supplement 1C* and D, cells were cultured in the medium supplemented with 25 µM Thiamet-G (Selleck, s7213) or 25 µM OSMI-1(Sigma, SML1621) for 6 hr before lysis. For the experiment in *Figure 4D*, the gel was stained with a Fast Silver Stain Kit (Beyotime, P0017S).

## Immunoprecipitation

For the immunoprecipitation experiment in *Figures 2C and 3B*, the HEK293T cells ($1 \times 10^7$ cells per sample) and flies (~20 flies per sample) were lysed in RIPA lysis buffer (50 mM Tris pH 8.0, 150 mM NaCl, 0.1% SDS, 0.5% Sodium deoxycholate, 1% NP40, 10 mM NaF, 10 mM $Na_2VO_4$, 50 µM Thiamet-G) supplemented with protease inhibitor cocktail (1:100, Sigma, P8340) and PMSF (1:100, Sigma, P7626) on ice for 30 min. After centrifugation at 13,000 g for 30 min at 4°C, the supernatants were transferred to new tubes. The protein concentration was determined using a BCA assay (Beyotime, p0009). Streptavidin magnetic beads (MCE, HY-K0208) were washed twice with RIPA lysis buffer, and incubated with the same amount of lysate from TurboID-*Cp*OGA$^{CD}$ or control samples on a rotator overnight at 4°C. The beads were washed twice with 1 mL of RIPA lysis buffer, once with 1 mL of 1 M KCl, once with 1 mL of 0.1 M $Na_2CO_3$, once with 1 mL of 2 M urea in 10 mM Tris-HCl (pH 8.0), and twice with 1 mL RIPA lysis buffer. After that, the beads were resuspended in SDS sample buffer and boiled for 10 min at 95°C. Finally, samples were stored at −80°C for future analysis.

The immunoprecipitation experiment in *Figure 4B* was performed as previously described (*Huang et al., 2019*). Briefly, fly brains (~40 fly brains per sample) were lysed in ribo-lysis buffer (50 mM Tris-HCl pH 7.4, 12 mM $MgCl_2$, 100 mM KCl, 1 mM DTT, 1% NP-40, 100 µg/mL cycloheximide, 50 µM Thiamet-G) supplemented with protease inhibitor cocktail (1:100, Sigma, P8340) and PMSF (1:100, Sigma, P7626) on ice for 30 min. After centrifugation at 13,000 g for 30 min at 4°C, the supernatants were transferred to new tubes. The protein concentration was determined using a BCA assay (Beyotime, p0009). Anti-FLAG M2 affinity gels (Sigma, A2220) were washed twice with ribo-lysis buffer, and incubated with tissue lysates on a rotator overnight at 4°C. The beads were washed three times with 1 mL of high salt buffer (50 mM Tris-HCl pH 7.4, 12 mM $MgCl_2$, 300 mM KCl, 1 mM DTT, 1% NP-40, 100 µg/mL cycloheximide). The beads were resuspended in SDS sample buffer and boiled for 10 min at 95°C. Finally, samples were stored at −80°C for future analysis.

## Immunofluorescence

The adult fly brains were dissected in PBS and fixed with 4% paraformaldehyde (PFA, Biosharp, BL539A) for 1 hr at room temperature. The brains were washed three times with PBS (Biological Industries, 02-023-1A) and then permeabilized and blocked in 5% BSA (Biofroxx, 4240GR005) in 0.3% PBST (PBS with 0.3% Triton X-100) for 90 min at room temperature. After being washed three times with PBS, the brains were incubated with primary antibodies overnight at 4°C, washed three times with PBS, and incubated with secondary antibodies (1:200, Thermo Fisher Scientific) and DAPI (1:500, Sigma, D9542) for 1 hr at room temperature. The brains were then washed three times with PBS and imaged by confocal fluorescence microscopy (Zeiss LSM880) with a 20x objective. Z-stacks were acquired with a spacing of 1 µm. Primary antibodies were dissolved in 5% BSA (Biofroxx, 4240GR005) and the dilutions were: Streptavidin-Cy3 (1:200, BioLegend, 405215), RL2 (1:200, Abcam, ab2739), HA (1:200, Cell Signaling Technology, 3724), and GFP (1:200, Cell Signaling Technology, 2955).

## Measurement of protein synthesis

The protein synthesis in fly brains was assessed using the Click-iT Plus OPP Alexa Fluor 594 Protein Synthesis Assay Kit (Thermo Fisher Scientific, C10457). Fly brains were dissected in *Drosophila* medium (Gibco, 21720024) and then incubated in a medium containing 1:1000 (20 µM) of Click-iT OPP reagent at room temperature for 30 min. The brains were washed three times with PBS, and then fixed with 4% PFA (Biosharp, BL539A) for 1 hr at room temperature. The brains were permeabilized and blocked in 5% BSA (Biofroxx, 4240GR005) in 0.3% PBST (PBS with 0.3% Triton X-100) for 90 min at room temperature, and then washed three times with PBS. The brains were incubated with primary anti-bodies (GFP, 1:200, Cell Signaling Technology, 2955) overnight at 4°C, washed three times with PBS, and incubated with secondary antibodies (1:200, Thermo Fisher Scientific) and DAPI (1:500, Sigma, D9542) for 1 hr at room temperature. For the Click-iT reaction, brains were incubated in the Click-iT reaction cocktail in the dark at room temperature for 30 min. Brains were then washed three times with PBS and imaged by confocal fluorescence microscopy (ZEISS LSM880).

## RT-qPCR

RNA was extracted from flies using TRIzol (Life Technologies, 87804), and 1 µg total RNA was reverse transcribed to generate cDNA using RevertAid First Strand cDNA Synthesis Kit (Thermo Fisher Scientific, K1621). The cDNA was then used as templates and qPCR was performed using the SYBR Green qPCR Master Mix (Solomon Biotech, QST-100) on the QuantStudio3 Real-Time PCR system (Applied Biosystems). The expression levels for each gene were normalized to Actin. Detailed information about the primers was listed in *Supplementary file 9*.

## Protein identification by LC-MS/MS

The HEK293T cells ($2×10^7$ cells per sample) and fly brains (~200 fly brains that expressed TurboID-*Cp*OGA$^{CD/DM}$ in brain neurons per sample, ~800 fly brains that expressed TurboID-*Cp*OGA$^{CD/DM}$ in other brain structures per sample, three biological replicates) were immunoprecipitated with strepta-vidin magnetic beads as described above. The supernatants were used for SDS-PAGE separation and minimally stained with Coomassie brilliant blue (Solarbio, C8430-10g). The gels were cut into small pieces, and reduced and alkylated in 10 mM DTT and 55 mM IAA (Merck, I6125), respectively. For digestion, 0.5 µg sequencing-grade modified trypsin was added and incubated at 37°C overnight. The peptides were then collected, desalted by StageTip (Thermo Fisher Scientific, 87782), and resolved in 0.1% formic acid before analysis by mass spectrometry. Mass spectrometry analysis was performed using Q Exactive HF-X mass spectrometer (Thermo Fisher Scientific) coupled with Easy-nLC 1200 system. Mobile phases A and B were water and 80% acetonitrile, respectively, with 0.1% formic acid. Protein digests were loaded directly onto an analytical column (75 µm×15 cm, 1.9 µm C18, 1 µm tip) at a flow rate of 450 nL/min. Data were collected in a data-dependent manner using a top 25 method with a full MS mass range from 400 to 1400 m/z, 60,000 resolutions, and an AGC target of $3×10^6$. MS2 scans were triggered when an ion intensity threshold of $4×10^5$ was reached. A dynamic exclusion time of 30 s was used. Ions with charge state 6–8 and more than eight were excluded.

## Data analysis

The raw data were imported into the MaxQuant software to identify and quantify the proteins. The following parameters were used: trypsin for enzyme digestion; oxidation of methionine, acetylation of the protein N terminus, biotinylation of lysine and protein N terminus and HexNAc (ST) as variable modifications; carbamidomethyl (C) as fixed modification. We used the canonical human protein data-base (containing 20,379 reviewed protein isoforms) or *Drosophila melanogaster* protein database (containing 22,088 protein isoforms, including reviewed and unreviewed sequences) for database searching separately. The false discovery rate (FDR) was 1% for peptide-spectrum matches (PSM) and protein levels. For the proteomics data of different brain regions of *Drosophila*, we used label-free quantitation (LFQ) to determine the relative amounts of proteins among three replicates. Perseus software was used to filter out all contaminates identified by MaxQuant (contaminant proteins, reversed proteins, proteins only identified by site). A pseudocount of 1 was added to protein inten-sities in order to avoid taking the log of 0. We generated $\log_2$ Fold Change ($\log_2$ FC) values for each protein in the TurboID-*Cp*OGA$^{CD}$ group relative to the TurboID-*Cp*OGA$^{DM}$ control. For the proteomics data of HEK293T cell, only proteins identified with at least 2 peptides were considered for further

analysis. Proteins were considered as *O*-GlcNAcylated substrates when differences in $\log_2$ FC of TurboID-*Cp*OGA^CD group with relative to the TurboID-*Cp*OGA^DM control were higher than 1. For the proteomics data from different brain regions of *Drosophila*, only proteins identified with at least 2 peptides and in at least 2 of the 3 replicates of TurboID-*Cp*OGA^CD were included for further analysis. A two-tailed unpaired student's t-test was applied in order to determine the statistical significance of the differences. Proteins were considered as *O*-GlcNAcylated substrates when differences in $\log_2$ FC of TurboID-*Cp*OGA^CD group with relative to the TurboID-*Cp*OGA^DM control were higher than 1 or statistically significant (p<0.05).

To adjust the interference caused by varying abundance of the putative *O*-GlcNAc substrates in different brain regions, single-cell transcriptomic data of the entire adult *Drosophila* brain (GEO: GSE107451) (*Davie et al., 2018*) was used to generate a normalizing factor for each substrate. Briefly, the annotated cell clusters were categorized into different brain regions. Then, the average mRNA expression level of each gene within a certain brain region was calculated. The normalizing factor was defined as the ratio of the average mRNA expression level of a given gene in neurons from a specific brain region to the average mRNA expression level of the same gene in neurons from the whole brain (*Supplementary file 10*). The normalized *O*-GlcNAc level was generated as the *O*-GlcNAc level ($\log_2$ FC) of a putative *O*-GlcNAcylated protein divided by its normalizing factor in a certain brain region (*Supplementary file 11*).

### Website

The website was created to browse through the *O*-GlcNAc database (https://www.kyuanlab.com/tsOGA), using the database managem the ent system Centos and the uWSGI web framework. Backend servers were developed by Python programming language (version 3.7). GNU/Linux Debian-based systems with Gunicorn (Python HTTP) and NginX were used for the development and production of the website. The website search function was based on MySQL database.

### Quantification and statistical analysis

To quantify fluorescent intensities in different *Drosophila* brain regions, whole brain images were stitched together using the stitching algorithm in ZEN software (Zeiss), and maximum intensity projection was produced. The images were then analyzed using ImageJ software. Mean fluorescent intensity of the whole brain or ROI was measured, and the relative fluorescent intensity was calculated as a ratio of the mean fluorescent intensity in ROI to that of the whole brain.

GO enrichment analyses of *O*-GlcNAcome in HEK293T cells and *Drosophila* were performed using *DAVID*. Protein-protein interaction (PPI) network of *O*-GlcNAcome in HEK293T cells was performed using *STRING*. GraphPad Prism was used for statistical analysis and the student's t-test was used to determine statistical significance. Bubble plots, pie plots and bar graphs were created using *Hiplot*, venn plots were created using *jvenn*.

### Materials availability

All cells and fly strains generated in this study are available upon request to the lead contact (see above).

### Lead contact

Further information and requests for resources and reagents should be directed to and will be fulfilled by the lead contact, Dr. Kai Yuan (yuankai@csu.edu.cn).

## Acknowledgements

We gratefully acknowledge Drs. Jilong Liu, Hai Huang, Feng He, Yan Chen, Pishun Li, Ranhui Duan, the Developmental Studies Hybridoma Bank, the Bloomington *Drosophila* Stock Center, and TsingHua Fly Center for reagents and fly stocks. We thank colleagues in the center for medical genetics and members of the Yuan lab for helpful discussions.

# Additional information

## Funding

| Funder | Grant reference number | Author |
|---|---|---|
| National Natural Science Foundation of China | 92153301 | Kai Yuan |
| National Natural Science Foundation of China | 91853108 | Kai Yuan |
| National Natural Science Foundation of China | 32170821 | Kai Yuan |
| National Natural Science Foundation of China | 32101034 | Fang Chen |
| Department of Science and Technology of Hunan Province | 2021JJ10054 | Kai Yuan |
| Department of Science and Technology of Hunan Province | 2019SK1012 | Kai Yuan |
| Central South University | 2021zzts0566 | Haibin Yu |
| Central South University | 2019zzts046 | Yaowen Zhang |
| Central South University | 2020CX016 | Kai Yuan |
| Villum Fonden | 00054496 | Daan MF van Aalten |
| Novo Nordisk Fonden | NNF21OC0065969 | Daan MF van Aalten |

The funders had no role in study design, data collection and interpretation, or the decision to submit the work for publication.

## Author contributions

Haibin Yu, Data curation, Software, Formal analysis, Validation, Investigation, Methodology, Writing - original draft; Dandan Liu, Data curation, Investigation, Methodology; Yaowen Zhang, Formal analysis, Validation, Investigation, Visualization, Methodology; Ruijun Tang, Xunan Fan, Lu Lv, Validation; Song Mao, Data curation, Software, Investigation, Visualization, Methodology; Fang Chen, Validation, Methodology; Hongtao Qin, Bing Yang, Resources, Methodology; Zhuohua Zhang, Resources, Supervision; Daan MF van Aalten, Conceptualization, Resources, Writing – review and editing; Kai Yuan, Conceptualization, Resources, Formal analysis, Supervision, Funding acquisition, Investigation, Visualization, Methodology, Project administration, Writing – review and editing

## Author ORCIDs

Haibin Yu http://orcid.org/0000-0002-8002-1316
Fang Chen http://orcid.org/0009-0000-6521-6580
Kai Yuan http://orcid.org/0000-0001-7002-5703

## Decision letter and Author response

Decision letter https://doi.org/10.7554/eLife.91269.sa1
Author response https://doi.org/10.7554/eLife.91269.sa2

# Additional files

## Supplementary files

• Supplementary file 1. *O*-GlcNAcylated proteins identified by TurboID-*Cp*OGA$^{CD}$ from HEK293T cells.

• Supplementary file 2. Previously identified *O*-GlcNAcylated proteins from HEK293T cells, related to *Figure 2D* and *Figure 2—figure supplement 1E*.

• Supplementary file 3. 48 proteins shared among the three datasets, related to *Figure 2D* and

*Figure 2—figure supplement 1E.*

• Supplementary file 4. *O*-GlcNAcylated proteins identified by TurboID-*Cp*OGA<sup>CD</sup> from brain neuron of *Drosophila*.

• Supplementary file 5. *O*-GlcNAcylated proteins identified by TurboID-*Cp*OGA<sup>CD</sup> from mushroom body of *Drosophila*.

• Supplementary file 6. *O*-GlcNAcylated proteins identified by TurboID-*Cp*OGA<sup>CD</sup> from antennal lobe of *Drosophila*.

• Supplementary file 7. *O*-GlcNAcylated proteins identified by TurboID-*Cp*OGA<sup>CD</sup> from ellipsoid body of *Drosophila*.

• Supplementary file 8. *O*-GlcNAcylated proteins identified by TurboID-*Cp*OGA<sup>CD</sup> from optic lobe of *Drosophila*.

• Supplementary file 9. Sequences of all the primers used in this study.

• Supplementary file 10. Cell clusters in different brain regions generated from single-cell transcriptomic data.

• Supplementary file 11. The normalized *O*-GlcNAc levels of *O*-GlcNAcylated proteins in different brain regions.

• MDAR checklist

## Data availability

The accession numbers for the mass spectrometry data were PXD040547 and PXD040412 on the Proteome X change Consortium PRIDE partner repository. All data generated during this study are included in the manuscript and supporting file; Source Data files have been uploaded to Dryad (https://doi.org/10.5061/dryad.sj3tx969t).

The following datasets were generated:

| Author(s) | Year | Dataset title | Dataset URL | Database and Identifier |
|---|---|---|---|---|
| Haibin Y | 2024 | Tissue-specific O-GlcNAcylation profiling reveals enrichment of ribosomal substrates in *Drosophila* mushroom body critical for associative learning | https://www.ebi.ac.uk/pride/archive/projects/PXD040547 | PRIDE, PXD040547 |
| Haibin Y | 2024 | *Drosophila* brain O-GlcNAc proteome | https://www.ebi.ac.uk/pride/archive/projects/PXD040412 | PRIDE, PXD040412 |
| Haibin Y | 2024 | Dataset-Tissue-specific O-GlcNAcylation profiling identifies substrates in translational machinery in *Drosophila* mushroom body contributing to olfactory learning | https://doi.org/10.5061/dryad.sj3tx969t | Dryad Digital Repository, 10.5061/dryad.sj3tx969t |

The following previously published datasets were used:

| Author(s) | Year | Dataset title | Dataset URL | Database and Identifier |
|---|---|---|---|---|
| Davie K, Janssens J, Koldere D, De Waegeneer M, Pech U, Kreft L, Aibar S, Makhzami S, Christiaens V, Bravo Gonzalez-Blas C | 2018 | A single-cell transcriptome atlas of the ageing *Drosophila* brain | https://www.ncbi.nlm.nih.gov/geo/query/acc.cgi?acc=GSE107451 | NCBI Gene Expression Omnibus, GSE107451 |

*Continued on next page*

*Continued*

| Author(s) | Year | Dataset title | Dataset URL | Database and Identifier |
|---|---|---|---|---|
| Hahne H, Sobotzki N, Nyberg T, Helm D, Borodkin VS, van Aalten DM, Agnew B, Kuster B | 2013 | Proteome-wide purification of O-GlcNAc proteins | https://www.ebi.ac. uk/pride/archive/ projects/PXD000061 | PRIDE, PXD000061 |

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

# Appendix 1

### Appendix 1—key resources table

| Reagent type (species) or resource | Designation | Source or reference | Identifiers | Additional information |
|---|---|---|---|---|
| Genetic reagent (*Drosophila melanogaster*) | UAS-HA-TurboID-CpOGA<sup>CD</sup> | This paper | | Expresses HA-TurboID-*Cp*OGA<sup>CD</sup> under the control of UAS. |
| Genetic reagent (*Drosophila melanogaster*) | UAS-HA-TurboID-CpOGA<sup>DM</sup> | This paper | | Expresses HA-TurboID-*Cp*OGA<sup>DM</sup> under the control of UAS. |
| Genetic reagent (*Drosophila melanogaster*) | UAS-GFP-CpOGA<sup>WT</sup> | This paper | | Expresses GFP-*Cp*OGA<sup>WT</sup> under the control of UAS. |
| Genetic reagent (*Drosophila melanogaster*) | UAS-GFP-CpOGA<sup>DM</sup> | This paper | | Expresses GFP-*Cp*OGA<sup>DM</sup> under the control of UAS. |
| Genetic reagent (*Drosophila melanogaster*) | Da-Gal4 | Bloomington *Drosophila* Stock Center | BDSC, #95282 | w[*]; P{w[+mW.hs]=GAL4 da.G32}2; P{w[+mW.hs]=GAL4 da.G32}UH1 |
| Genetic reagent (*Drosophila melanogaster*) | Elav-Gal4 | Bloomington *Drosophila* Stock Center | BDSC, #8765 | P{w[+mC]=GAL4 elav.L}2/CyO |
| Genetic reagent (*Drosophila melanogaster*) | OK107-Gal4 | Bloomington *Drosophila* Stock Center | BDSC, #854 | w[*]; P{w[+mW.hs]=GawB} OK107 ey[OK107]/In(4)ci[D], ci[D] pan[ciD] sv[spa-pol] |
| Genetic reagent (*Drosophila melanogaster*) | 201Y-Gal4 | Bloomington *Drosophila* Stock Center | BDSC, #4440 | w[1118]; P{w[+mW.hs]=GawB} Tab2[201Y] |
| Genetic reagent (*Drosophila melanogaster*) | C232-Gal4 | Bloomington *Drosophila* Stock Center | BDSC, #30828 | w[*]; P{w[+mW.hs]=GawB} Alp4[c232] |
| Genetic reagent (*Drosophila melanogaster*) | GMR14H04-Gal4 | Bloomington *Drosophila* Stock Center | BDSC, #48655 | w[1118]; P{y[+t7.7] w[+mC]=GMR14 H04-GAL4} attP2 |
| Genetic reagent (*Drosophila melanogaster*) | GMR33H10-Gal4 | Bloomington *Drosophila* Stock Center | BDSC, #49762 | w[1118]; P{y[+t7.7] w[+mC]=GMR33 H10-GAL4} attP2 |
| Genetic reagent (*Drosophila melanogaster*) | uas-RPL13A-FLAG | Bloomington *Drosophila* Stock Center | BDSC, #83684 | w[*]; P{w[+mC]=UAS-RpL13A. FLAG}3 |
| Genetic reagent (*Drosophila melanogaster*) | uas-dMyc | Bloomington *Drosophila* Stock Center | BDSC, #9674 | w[1118]; P{w[+mC]=UAS Myc.Z}132 |

*Appendix 1 Continued on next page*

*Appendix 1 Continued*

| Reagent type (species) or resource | Designation | Source or reference | Identifiers | Additional information |
|---|---|---|---|---|
| Genetic reagent (*Drosophila melanogaster*) | uas-shLuciferase | Bloomington *Drosophila* Stock Center | BDSC, #31603 | y[1] v[1]; P{y[+t7.7] v[+t1.8]=TRiP.JF01355}attP2 |
| Genetic reagent (*Drosophila melanogaster*) | uas-shRPL5 | TsingHua Fly Center (THFC) | THU0670 | |
| Genetic reagent (*Drosophila melanogaster*) | uas-shRPs26 | TsingHua Fly Center (THFC) | THU0747 | |
| Genetic reagent (*Drosophila melanogaster*) | uas-shRPL24 | TsingHua Fly Center (THFC) | THU1411 | |
| Genetic reagent (*Drosophila melanogaster*) | uas-shRPS6 | TsingHua Fly Center (THFC) | THU0864 | |
| Genetic reagent (*Drosophila melanogaster*) | uas-shRPL11 | TsingHua Fly Center (THFC) | TH201500769.S | |
| Genetic reagent (*Drosophila melanogaster*) | uas-shRPS3 | TsingHua Fly Center (THFC) | THU1958 | |
| Genetic reagent (*Drosophila melanogaster*) | uas-shRPL32 | TsingHua Fly Center (THFC) | TH201500773.S | |
| Genetic reagent (*Drosophila melanogaster*) | uas-shRPS28b | TsingHua Fly Center (THFC) | THU1037 | |
| Cell line (*Homo sapiens*) | HEK293T cells | Meisen CTCC | Cat# CTCC-001–0188 | Procured from ATCC (CRL-3216) |
| Antibody | Anti-O-Linked N-Acetylglucosamine Antibody, Mouse Monoclonal, RL2 | Abcam | Cat# ab2739, RRID: AB_30326 | WB (1:1000) IF (1:200) |
| Other | Cyanine3 Streptavidin | BioLegend | Cat# 405215 | IF (1:200) |
| Other | Streptavidin HRP | GenScript | Cat# M00091 | WB (1:2000) |
| Antibody | Anti-HA-Tag Rabbit Monoclonal Antibody (C29F4) | Cell Signaling Technology | Cat# 3724, RRID: AB_1549585 | WB (1:3000) IF (1:200) |
| Antibody | Anti-GFP Mouse Monoclonal Antibody (4B10) | Cell Signaling Technology | Cat# 2955, RRID: AB_1196614 | WB (1:1000) IF (1:200) |
| Antibody | DYKDDDDK Tag (D6W5B) Rabbit Monoclonal (Anti-FLAG M2 Antibody) | Cell Signaling Technology | Cat# 14793, RRID: AB_2572291 | WB (1:3000) |

*Appendix 1 Continued on next page*

*Appendix 1 Continued*

| Reagent type (species) or resource | Designation | Source or reference | Identifiers | Additional information |
|---|---|---|---|---|
| Antibody | Anti-α-Tubulin, Mouse Monoclonal, HRP Conjugate (DM1A) | Cell Signaling Technology | Cat# 12351 S, RRID: AB_2797891 | WB (1:3000) |
| Antibody | Goat Anti-Rabbit IgG (H+L) Secondary Antibody, Polyclonal secondary, Alexa Fluor-488 | Thermo Fisher Scientific | Cat# A-31565, RRID: AB_2536178 | IF (1:200) |
| Antibody | Goat Anti-Mouse IgG (H+L) Secondary Antibody, Polyclonal secondary, Alexa Fluor-488 | Thermo Fisher Scientific | Cat# A32723, RRID: AB_2633275 | IF (1:200) |
| Antibody | Goat Anti-Mouse IgG (H+L) Secondary Antibody, Polyclonal secondary, Alexa Fluor-546 | Thermo Fisher Scientific | Cat# A-11030, RRID: AB_2534089 | IF (1:200) |
| Antibody | Goat anti-Mouse IgG (H+L) Secondary Antibody, Polyclonal secondary | Thermo Fisher Scientific | Cat# 31160, RRID: AB_228297 | WB (1:5000) |
| Antibody | Goat anti-Rabbit IgG (H+L) Secondary Antibody, Polyclonal secondary | Thermo Fisher Scientific | Cat# A16098, RRID: AB_2534772 | WB (1:5000) |
| Commercial assay or kit | Click-iT Plus OPP Alexa Fluor 594 Protein Synthesis Assay Kit | Thermo Fisher Scientific | Cat# C10457 | |
| Commercial assay or kit | cDNA using RevertAid First Strand cDNA Synthesis Kit | Thermo Fisher Scientific | Cat# K1621 | |
| Commercial assay or kit | Fast Silver Stain Kit | Beyotime | Cat# P0017S | |
| Chemical compound, drug | DAPI | Sigma | Cat# D9542 | |
| Chemical compound, drug | Biotin | Merck | Cat# B4501 | |
| Chemical compound, drug | 3-Octanol (OCT) | Sigma | Cat# 218405 | |
| Chemical compound, drug | trans-4-Methylcyclohexanol (MCH) | Sigma | Cat# 104191 | |
| Chemical compound, drug | Protease Inhibitor Cocktail | Sigma | Cat# P8340 | |
| Chemical compound, drug | Phenylmethanesulfonyl fluoride (PMSF) | Sigma | Cat# P7626 | |
| Chemical compound, drug | Thiamet-G | Selleck | Cat# s7213 | |
| Chemical compound, drug | OSMI-1 | Sigma | Cat# SML1621 | |

*Appendix 1 Continued on next page*

*Appendix 1 Continued*

| Reagent type (species) or resource | Designation | Source or reference | Identifiers | Additional information |
|---|---|---|---|---|
| Chemical compound, drug | Streptavidin Magnetic Beads | MCE | Cat# HY-K0208 | |
| Chemical compound, drug | Anti-FLAG Affinity Gel | Sigma | Cat# A2220 | |
| Chemical compound, drug | SYBR Green qPCR Master Mix | SolomonBio | Cat# QST-100 | |
| Chemical compound, drug | Sequencing-grade modified trypsin | Promega | Cat# V5111 | |
| Chemical compound, drug | α-Iodoacetamide (IAA) | Merck | Cat# I6125 | |
| Software, algorithm | MaxQuant | Max Planck Institute of Biochemistry | https://www.maxquant.org | |
| Software, algorithm | Perseus | Max Planck Institute of Biochemistry | https://maxquant.net/perseus/ | |
| Software, algorithm | GraphPad Prism 8 | GraphPad Software | https://www.graphpad.com/scientificsoftware/prism/ | |
| Software, algorithm | Fiji | ImageJ | http://fiji.sc/ | |
| Software, algorithm | Python | N/A | https://www.python.org/ | |
| Software, algorithm | Illustrator | Adobe | https://www.adobe.com/uk/products/illustrator.html | |
| Software, algorithm | Zeiss ZEN 2.3 (blue edition) | Carl Zeiss Microscopy GmbH | https://www.zeiss.com/microscopy/int/products/microscopesoftware/zen.html | |

