## [Editor Report]

This valuable study provides solid evidence that within the *Drosophila* brain there are regionally regulated patterns of O-linked modification of proteins with the monosaccharide N-Acetyl glucosamine. Using a novel and powerful method of identifying proteins bearing this modification, the authors provide evidence that in a region of the *Drosophila* brain critical for associative learning, the mushroom body, there is a high representation of modified proteins affecting protein translation. Reductions in GlcNAc modification affects both an associative learning task and new protein synthesis, suggesting a critical function of these monosaccharide modifications in the regulation of protein synthesis required for memory formation. These findings provide a putative mechanism for human neurological deficits that accompany reductions in this ubiquitous carbohydrate modification.

---

## [Decision Letter]

**Decision letter after peer review:**

Thank you for submitting your article "Olfactory learning in *Drosophila* requires *O-*GlcNAcylation of mushroom body ribosomal subunits" for consideration by *eLife*. Your article has been reviewed by 2 peer reviewers, and the evaluation has been overseen by a Reviewing Editor and K VijayRaghavan as the Senior Editor.

Essential revisions (for the authors):

1) Potential reasons for the very low overlap with other methods to identify GlycNac-modified proteins requires more discussion o, as well as potential problems with the authors own approach

2) The current data do not rule out the possibility that altering GLcNAC affects LTM through a mechanism independent of ribosome modification and/or activity dependent translational control. It should be clearly acknowledged in the abstract and discussion that the data fall short of causally connecting ribosomal protein modification with translational control required for LTM. Phrases like "consistent with" may be useful.

3) Please respond carefully and as you consider appropriate all of the other comments of the reviewers.

*Reviewer #1 (Recommendations for the authors):*

1. This study relies almost exclusively on the over-expression of foreign proteins and interpreting their effects based solely on their known affinity and activity for O-GlcNac modifications. The effects of high levels of over-expression are often hard to identify or know and while there are appropriate controls in their studies, using catalytically inactive mutant forms for comparison, there is no discussion of this issue. The authors do describe that over-expression in optic and antennal lobes for example, provided reason for exclusion by virtue of differences in susceptibility (that term is not explained), indicating there are behavioral effects that can interfere with their assays. Some direct discussion of this potential problem is warranted.

2. Their comparison of several methods of identifying O-GlcNAc-modified proteins was tested in HEK293 cells, providing a measure of their method of proximity-labelling using their Clostridium OGA construct. The set of proteins identified by all three tested methods revealed 48, a modest number compared to the total sets for each method (2119, 273 and 336) for chemoenzymatic, direct capture and TurboID-CpOGA. This level of concordance suggests that different methods might well identify different sets of proteins, a concern for interpreting the functional significance of these changes. Do the authors argue that these results only affect sensitivity or could it possibly bias the classes of O-GlcNAc modified proteins identified?

3. The rescue of associative learning phenotypes mediated by reductions in O-GlcNAc modification by over-expression of dMyc is a very important experiment that supports their conclusion that translational deficits are the basis of the learning defects. The authors should explain further in the text the rationale and basis for this experiment, namely, how dMyc affects the protein synthetic machinery and why these changes could rescue loss of O-GlcNAc modifications.

4. The paper is challenging to read, and I suggest the authors provide it for reading and editing by someone not in their immediate field in order to make changes that make it more accessible to a general reader.

5. Their graphics are terrific and provide great assistance in clarifying the figures. Well done.

*Reviewer #2 (Recommendations for the authors):*

As it stands the manuscript does not support the idea that O-GlcNAcylation of ribosomal proteins in the mushroom body (MB) is required for protein synthesis and olfactory learning. Experiments targeting specific ribosomal components could be a compelling demonstration. For example, Ser/Thr residues that get O-GlcNAcylated in RPL24 or RPS3 could be mutated. If these mutants showed learning defects it would be an important result. Same type of mutants in ex vivo OPP assays could be performed to determine their effect on protein synthesis. These types of experiments would definitely show specific requirement of ribosomal protein modification for ribosomal activity and associative learning.

---

## [Author Response]

Essential revisions (for the authors):1) Potential reasons for the very low overlap with other methods to identify GlycNac-modified proteins requires more discussion o, as well as potential problems with the authors own approach

Thank you for raising this important issue. A lot of effort has been made to identify protein *O-*GlcNAc modifications and many profiling methods have been established in the past 30 years^1^. However, as you pointed out, the *O*-GlcNAcome identified using different methods are quite diverse. There are at least two possible reasons behind the observed low overlap.

First, *O*-GlcNAcylation is highly responsive to a wide range of intrinsic and extrinsic stimuli, acting as an important cellular metabolic and stress sensing mechanism^2^. Meanwhile, as a monosaccharide modification, the addition and removal of *O-*GlcNAc moiety can be very rapid, with cycling rates as short as several minutes^3,4^. This dynamic nature of *O*-GlcNAc cycling contributes to the observed diversity among different profiling attempts.

Second, the high lability of the GlcNAc moiety and low abundance of the *O*-GlcNAcylated proteins pose significant challenges in direct utilization of mass spectrometry for *O*-GlcNAcylation profiling^1,5,6^. Different enrichment methods are needed to protect and stabilize the GlcNAc moiety. This unavoidably introduces bias in substrates preference intrinsic to the methods, which also contributes to the limited overlap of the identified *O*-GlcNAc substrates.

As for our approach reported here, there are several potential caveats: (1) Similar to the direct capture methods, the sensitivity is limited compared to the chemoenzymatic labeling methods. Additionally, our method uses the differential enrichment in the TurboID-*Cp*OGA^CD^ experimental group relative to the TurboID-*Cp*OGA^DM^ control group to identify putative *O*-GlcNAc substrates. Therefore, the sensitivity is also influenced by the cutoff chosen in the analysis. (2) Our method relies on ectopic expression of bacterial *Cp*OGA mutants fused with TurboID to label the *O*-GlcNAcome. The introduction of these foreign proteins could impose stress on the targeted tissue and interfere with its normal functions. Therefore, we included *Cp*OGA^DM^ as a control in all the experiments to minimize possible artifacts. (3) The TurboID used in our method can biotinylate all the proximal proteins within ~ 10 nm radius. As a result, the identified proteins can be in complex with other *O*-GlcNAc substrates but itself is not directly *O*-GlcNAcylated. To be accurate, we used “putative” or “candidate” substrates and *O*-GlcNAc “interactome” in our description.

We have updated the Introduction and Discussion to include these aforementioned points.

2) The current data do not rule out the possibility that altering GlcNAc affects LTM through a mechanism independent of ribosome modification and/or activity dependent translational control. It should be clearly acknowledged in the abstract and discussion that the data fall short of causally connecting ribosomal protein modification with translational control required for LTM. Phrases like "consistent with" may be useful.

We appreciate that you bring up this crucial point. Our study demonstrated that: (1) hypo-*O-*GlcNAcylation resulted in learning defect and reduced protein synthesis in mushroom body; (2) knockdown of several ribosomal subunits to reduce protein synthesis in mushroom body was sufficient to drive learning defect; (3) upregulation of protein synthesis by overexpression of dMyc in mushroom body ameliorated the cognitive defect caused by hypo-*O-*GlcNAcylation. These observations consolidated a link between hypo-*O-*GlcNAcylation and cognitive impairment in *Drosophila*, and suggested that insufficient *O-*GlcNAcylation in the mushroom body of *Drosophila* brain reduced local translational activity that contributed to the observed olfactory learning deficits. However, our data indeed fall short for a causal connection between *O*-GlcNAcylation of ribosomes and translational activity. In fact, our profiling also identified many translational initiation factors whose *O*-GlcNAcylation could impact translation as well.

Therefore, your point has been well taken, and we have made the following changes accordingly: (1) we changed the Title of our manuscript to put more emphasis on the method itself; (2) we replaced “ribosomal activity” with “translational activity” in all the texts to describe the findings more accurately; (3) we used words like “suggest” and “consistent with” to emphasize it is one of the explanations whenever possible; (4) we specifically pointed out in the Discussion that the impact of *O*-GlcNAcylation on ribosomal activity needs future structural and biochemical characterizations.

We think these changes have greatly improved the accuracy of our description, and we want to express our gratitude again for this suggestion.

3) Please respond carefully and as you consider appropriate all of the other comments of the reviewers.

We have prepared a point-by-point response to all the comments, and we express our sincere appreciation to you for the meticulous evaluation of our manuscript.

Reviewer #1 (Recommendations for the authors):1. This study relies almost exclusively on the over-expression of foreign proteins and interpreting their effects based solely on their known affinity and activity for O-GlcNAc modifications. The effects of high levels of over-expression are often hard to identify or know and while there are appropriate controls in their studies, using catalytically inactive mutant forms for comparison, there is no discussion of this issue. The authors do describe that over-expression in optic and antennal lobes for example, provided reason for exclusion by virtue of differences in susceptibility (that term is not explained), indicating there are behavioral effects that can interfere with their assays. Some direct discussion of this potential problem is warranted.

Thank you for this valuable advice. The reason we chose a bacterial OGA to manipulate *O*-GlcNAcylation levels in flies are following.

First, the function of OGA is highly conserved during evolution (Author response image 1). The N-terminal *O*-GlcNAc hydrolase catalytic domain of *Cp*OGA has sequence homology to *Drosophila* OGA^7,8^. Previous studies have shown that *Cp*OGA exhibits remarkable catalytic activity on *Drosophila O*-GlcNAcylated proteins^9^. We have codon-optimized *CpOGA* sequence to *Drosophila*, and constructed the transgenic flies.

Second, the first insight into the structure of OGA catalytic domain came from the crystal structure of *Cp*OGA^10^. Until now the structural information of *Drosophila* OGA is not available. The crystal structure of *Cp*OGA clearly reveals that the Asp298 residue is crucial for its catalytic activity, and Asp401 residue for *O-*GlcNAc peptide binding. This structural and mechanistic information can guide the rational design of active or control OGA, as well as the TurboID-fused OGA mutants for profiling, during the generation of transgenic lines.

Third, in *OGT* mutated patient-derived fibroblasts, the expression of OGA is often compensatorily downregulated via mechanisms that are currently not fully understood. We decided to use a bacterial OGA to bypass the potential feedback between OGT and OGA. Indeed, we found that the expression of *Cp*OGA^WT^ in *Drosophila* reduced total *O-*GlcNAcylation level without altering the mRNA abundance of endogenous *oga* or *sxc*^11^.

**Author response image 1. sa2fig1:** Schematic of the domain structures of human OGA, *Drosophila* OGA, and *Cp*OGA. The *O*-GlcNAcase activity resides in the GH domain (blue). The C-terminal of *Drosophila* OGA harbors a putative histone acetyltransferase domain (HAT-like). Residues 31–618 of *Cp*OGA (corresponding to the GH domain plus an additional C-terminal) are used in this study.

As for the experiments in the optic and antennal lobes, we found ectopic expression of *Cp*OGA^WT^ in these brain regions altered odor acuity toward MCH or OCT when compared with *Cp*OGA^DM^ control. This difference in olfactory susceptibility invalidated the learning tests. Therefore, we excluded these flies from subsequent olfactory learning test. The reason why expression of *Cp*OGA^WT^ in optic and antennal lobes impacted olfactory perception is not clear. We think the multisensory nature of *Drosophila* behavior and perception might be involved. Perhaps there are functional interactions between olfactory processing neuronal circuit and visual as well as sensory neuronal circuits.

We have updated the Results and Discussion accordingly to accommodate the key points.

2. Their comparison of several methods of identifying *O*-GlcNAc-modified proteins was tested in HEK293 cells, providing a measure of their method of proximity-labelling using their *Clostridium* OGA construct. The set of proteins identified by all three tested methods revealed 48, a modest number compared to the total sets for each method (2119, 273 and 336) for chemoenzymatic, direct capture and TurboID-*Cp*OGA. This level of concordance suggests that different methods might well identify different sets of proteins, a concern for interpreting the functional significance of these changes. Do the authors argue that these results only affect sensitivity or could it possibly bias the classes of *O*-GlcNAc modified proteins identified?

Thank you for raising this important issue. The possible reasons for the low overlap among datasets generated by different profiling methods are elaborated in Essential revisions 1. We have also updated the Introduction and Discussion accordingly.

As for the question of whether it only affects sensitivity or introduces bias to different classes of substrates, we performed gene ontology (GO) analysis on these three datasets and found that they were enriched in similar biological processes (Figure S2E). Therefore, despite different enrichment strategies unavoidably introducing bias in substrate preference, we think the main classes of *O*-GlcNAcylated substrates from different profiling attempts remain comparable.

3. The rescue of associative learning phenotypes mediated by reductions in *O*-GlcNAc modification by over-expression of dMyc is a very important experiment that supports their conclusion that translational deficits are the basis of the learning defects. The authors should explain further in the text the rationale and basis for this experiment, namely, how dMyc affects the protein synthetic machinery and why these changes could rescue loss of *O*-GlcNAc modifications.

Thank you for highlighting this point. One well-recognized function of MYC is to serve as a direct regulator of ribosome biogenesis, promoting protein synthesis through transcriptional control of RNA and protein components of ribosomes, as well as factors involved in processing and nuclear export of these ribosomal subunits ^12-14^. MYC binds to sequences surrounding rDNA and directly promotes Pol I transcription throughout this region^15-17^. It also contributes to driving 5S transcription by Pol III^18^. These activities rely on the ability of MYC to both interact with the factors that promote Pol I recruitment to rDNA promoters and induce changes in local chromatin organization within rDNA repeats. Moreover, MYC directly enhances the transcription of key factors involved in rRNA transcription, such as UBF and components of the SL1 complex^16^. Furthermore, MYC boosts Pol II-mediated transcription of genes that encode ribosomal proteins^18-20^. The *Drosophila* dMyc, like its vertebrate homologs, activates numerous target genes associated with ribosome biogenesis and translation^12,21-24^.

We think dMyc boosts ribosome biogenesis at transcription level, which compensates the reduced translational activity caused by the insufficient post-translational *O*-GlcNAc modification. We have updated the RESULTS to better explain the rationale for this experimental design.

4. The paper is challenging to read, and I suggest the authors provide it for reading and editing by someone not in their immediate field in order to make changes that make it more accessible to a general reader.

Thank you for the suggestion. We have thoroughly edited the manuscript to make it more accessible to general readers.

5. Their graphics are terrific and provide great assistance in clarifying the figures. Well done.

Thank you very much for the encouragement.

Reviewer #2 (Recommendations for the authors):As it stands the manuscript does not support the idea that O-GlcNAcylation of ribosomal proteins in the mushroom body (MB) is required for protein synthesis and olfactory learning. Experiments targeting specific ribosomal components could be a compelling demonstration. For example, Ser/Thr residues that get O-GlcNAcylated in RPL24 or RPS3 could be mutated. If these mutants showed learning defects it would be an important result. Same type of mutants in ex vivo OPP assays could be performed to determine their effect on protein synthesis. These types of experiments would definitely show specific requirement of ribosomal protein modification for ribosomal activity and associative learning.

We agree that the current data fall short for a direct connection between *O*-GlcNAcylation of ribosomes and translational activity. Therefore, we have modified the TITLE and thoroughly edited the descriptions throughout the manuscript to assure accuracy (see Essential revisions 2).

Because the whole translational machinery is heavily *O*-GlcNAcylated in the mushroom body, and our profiling method can not identify individual *O*-GlcNAcylation sites on the substrates, it is very difficult to design the mutational experiments as you suggested. Future biochemical and structural characterizations of ribosomes might reveal the key *O*-GlcNAcylation sites and verify the connection between *O*-GlcNAc modifications of ribosomes and their activity.

Reference

1. Dupas, T., Betus, C., Blangy-Letheule, A., Pele, T., Persello, A., Denis, M., and Lauzier, B. (2022). An overview of tools to decipher O-GlcNAcylation from historical approaches to new insights. Int J Biochem Cell Biol *151*, 106289. 10.1016/j.biocel.2022.106289.

2. Zachara, N.E., O'Donnell, N., Cheung, W.D., Mercer, J.J., Marth, J.D., and Hart, G.W. (2004). Dynamic O-GlcNAc modification of nucleocytoplasmic proteins in response to stress. A survival response of mammalian cells. J Biol Chem *279*, 30133-30142. 10.1074/jbc.M403773200.

3. Miller, M.W., Caracciolo, M.R., Berlin, W.K., and Hanover, J.A. (1999). Phosphorylation and glycosylation of nucleoporins. Arch Biochem Biophys *367*, 51-60. 10.1006/abbi.1999.1237.

4. Roquemore, E.P., Chevrier, M.R., Cotter, R.J., and Hart, G.W. (1996). Dynamic O-GlcNAcylation of the small heat shock protein α B-crystallin. Biochemistry *35*, 3578-3586. 10.1021/bi951918j.

5. Yin, R., Wang, X., Li, C., Gou, Y., Ma, X., Liu, Y., Peng, J., Wang, C., and Zhang, Y. (2021). Mass Spectrometry for O-GlcNAcylation. Front Chem *9*, 737093. 10.3389/fchem.2021.737093.

6. Ma, J., and Hart, G.W. (2014). O-GlcNAc profiling: from proteins to proteomes. Clin Proteomics *11*, 8. 10.1186/1559-0275-11-8.

7. Alonso, J., Schimpl, M., and van Aalten, D.M. (2014). O-GlcNAcase: promiscuous hexosaminidase or key regulator of O-GlcNAc signaling? J Biol Chem *289*, 34433-34439. 10.1074/jbc.R114.609198.

8. Muha, V., Fenckova, M., Ferenbach, A.T., Catinozzi, M., Eidhof, I., Storkebaum, E., Schenck, A., and van Aalten, D.M.F. (2020). O-GlcNAcase contributes to cognitive function in *Drosophila*. J Biol Chem *295*, 8636-8646. 10.1074/jbc.RA119.010312.

9. Mariappa, D., Selvan, N., Borodkin, V., Alonso, J., Ferenbach, A.T., Shepherd, C., Navratilova, I.H., and vanAalten, D.M.F. (2015). A mutant O-GlcNAcase as a probe to reveal global dynamics of protein O-GlcNAcylation during *Drosophila* embryonic development. Biochem J *470*, 255-262. 10.1042/BJ20150610.

10. Rao, F.V., Dorfmueller, H.C., Villa, F., Allwood, M., Eggleston, I.M., and van Aalten, D.M. (2006). Structural insights into the mechanism and inhibition of eukaryotic O-GlcNAc hydrolysis. EMBO J *25*, 1569-1578. 10.1038/sj.emboj.7601026.

11. Zhang, Y., Yu, H., Wang, D., Lei, X., Meng, Y., Zhang, N., Chen, F., Lv, L., Pan, Q., Qin, H., et al. (2023). Protein O-GlcNAcylation homeostasis regulates facultative heterochromatin to fine-tune sog-Dpp signaling during *Drosophila* early embryogenesis. J Genet Genomics. 10.1016/j.jgg.2023.05.014.

12. Gallant, P. (2013). Myc Function in *Drosophila*. Cold Spring Harbor Perspectives in Medicine *3*, a014324-a014324. 10.1101/cshperspect.a014324.

13. Jiao, L., Liu, Y., Yu, X.Y., Pan, X., Zhang, Y., Tu, J., Song, Y.H., and Li, Y. (2023). Ribosome biogenesis in disease: new players and therapeutic targets. Signal Transduct Target Ther *8*, 15. 10.1038/s41392-022-01285-4.

14. van Riggelen, J., Yetil, A., and Felsher, D.W. (2010). MYC as a regulator of ribosome biogenesis and protein synthesis. Nat Rev Cancer *10*, 301-309. 10.1038/nrc2819.

15. Arabi, A., Wu, S., Ridderstrale, K., Bierhoff, H., Shiue, C., Fatyol, K., Fahlen, S., Hydbring, P., Soderberg, O., Grummt, I., et al. (2005). c-Myc associates with ribosomal DNA and activates RNA polymerase I transcription. Nat Cell Biol *7*, 303-310. 10.1038/ncb1225.

16. Grandori, C., Gomez-Roman, N., Felton-Edkins, Z.A., Ngouenet, C., Galloway, D.A., Eisenman, R.N., and White, R.J. (2005). c-Myc binds to human ribosomal DNA and stimulates transcription of rRNA genes by RNA polymerase I. Nat Cell Biol *7*, 311-318. 10.1038/ncb1224.

17. Shiue, C.N., Berkson, R.G., and Wright, A.P. (2009). c-Myc induces changes in higher order rDNA structure on stimulation of quiescent cells. Oncogene *28*, 1833-1842. 10.1038/onc.2009.21.

18. Kawai, H., Kanegae, T., Christensen, S., Kiyosue, T., Sato, Y., Imaizumi, T., Kadota, A., and Wada, M. (2003). Responses of ferns to red light are mediated by an unconventional photoreceptor. Nature *421*, 287-290. 10.1038/nature01310.

19. Boon, K., Caron, H.N., van Asperen, R., Valentijn, L., Hermus, M.C., van Sluis, P., Roobeek, I., Weis, I., Voute, P.A., Schwab, M., and Versteeg, R. (2001). N-myc enhances the expression of a large set of genes functioning in ribosome biogenesis and protein synthesis. EMBO J *20*, 1383-1393. 10.1093/emboj/20.6.1383.

20. Popay, T.M., Wang, J., Adams, C.M., Howard, G.C., Codreanu, S.G., Sherrod, S.D., McLean, J.A., Thomas, L.R., Lorey, S.L., Machida, Y.J., et al. (2021). MYC regulates ribosome biogenesis and mitochondrial gene expression programs through its interaction with host cell factor-1. *ELife 10*. 10.7554/*eLife*.60191.

21. Orian, A., van Steensel, B., Delrow, J., Bussemaker, H.J., Li, L., Sawado, T., Williams, E., Loo, L.W., Cowley, S.M., Yost, C., et al. (2003). Genomic binding by the *Drosophila* Myc, Max, Mad/Mnt transcription factor network. Genes Dev *17*, 1101-1114. 10.1101/gad.1066903.

22. Grewal, S.S., Li, L., Orian, A., Eisenman, R.N., and Edgar, B.A. (2005). Myc-dependent regulation of ribosomal RNA synthesis during *Drosophila* development. Nature Cell Biology *7*, 295-302. 10.1038/ncb1223.

23. Hulf, T., Bellosta, P., Furrer, M., Steiger, D., Svensson, D., Barbour, A., and Gallant, P. (2005). Whole-genome analysis reveals a strong positional bias of conserved dMyc-dependent E-boxes. Mol Cell Biol *25*, 3401-3410. 10.1128/MCB.25.9.3401-3410.2005.

24. Marshall, L., Rideout, E.J., and Grewal, S.S. (2012). Nutrient/TOR-dependent regulation of RNA polymerase III controls tissue and organismal growth in *Drosophila*. EMBO J *31*, 1916-1930. 10.1038/emboj.2012.33.